# Dual targeting of CDK6 and LSD1 is synergistic and overcomes differentiation blockade in AML

Lise Brault [ID][1,2], Edwige Voisset [ID][1,2], Mathieu Desaunay [ID][1,2], Antonia Boudet[1,2], Paraskevi Kousteridou [ID][1,2,3], Sébastien Letard [ID][1,2], Nadine Carbuccia[1,2], Armelle Goubard[1,4], Rémy Castellano [ID][1,4], Yves Collette [ID][1,4,5], Julien Vernerey[1], Isabelle Vigon [ID][6], Jean-Max Pasquet[5,6], Patrice Dubreuil[1,2], Sophie Lopez [ID][1,2] & Paulo De Sepulveda [ID][1,2,5]✉

## Abstract

The heterogeneity of leukemic cells is the main cause of resistance to therapy in acute myeloid leukemia (AML). Consequently, innovative therapeutic approaches are critical to target a wide spectrum of leukemic clones, regardless of their genetic and non-genetic complexity. In this report, we leverage the vulnerability of AML cells to CDK6 to identify a combination therapy capable of targeting common biological processes shared by all leukemic cells, while sparing non-transformed cells. We demonstrate that the combined inhibition of CDK6 and LSD1 restores myeloid differentiation and depletes the leukemic progenitor compartment in AML samples. Mechanistically, this combination induces major changes in chromatin accessibility, leading to the transcription of differentiation genes and diminished LSC signatures. Remarkably, the combination is synergistic, induces durable changes in the cells, and is effective in PDX mouse models. While many AML samples exhibit only modest responses to LSD1 inhibition, co-targeting CDK6 restores the expected transcription response associated with LSD1 inhibition. Given the availability of clinical-grade CDK6 and LSD1 inhibitors, this combination holds significant potential for implementation in clinical settings through drug repositioning.

**Keywords** Leukemia; Kinase; Iadademstat; Palbociclib; Inhibitor
**Subject Categories** Cancer; Haematology

## Introduction

Resistance to current therapies is the main cause of disease relapse and death in acute myeloid leukemia (AML) (Shimony et al, 2023; Döhner et al, 2022). Cell heterogeneity and plasticity drive the persistence and selection of resistant leukemic cells. It is then crucial to uncover novel therapies capable of targeting most leukemic cells, including myeloid progenitor and stem cells with reconstitution capacity, irrespective of their mutational status.

Epigenetic dysregulation is a hallmark of AML resulting in the blockade of myeloid differentiation and enhanced leukemic stem cell features (Fennell et al, 2019). Indeed, the phenotype of leukemic cells is orchestrated by perturbed epigenetic and transcription factors activities, due to accumulated genetic alterations (Gutierrez and Romero-Oliva, 2013; Gallipoli et al, 2015). Furthermore, AML cells at relapse show convergent evolution of chromatin accessibility signatures. Among the epigenetic druggable targets in AML, lysine-specific histone demethylase 1 (LSD1) has gained attention due to its preferential overexpression in primary AML compared to normal hematopoietic stem and progenitor cells (Harris et al, 2012; Maes et al, 2018; Wingelhofer and Somervaille, 2019; Salamero et al, 2020). In leukemic cells, LSD1, together with the repressive CoREST complex, which includes HDAC1/2, is recruited by GFI1/GFI1B at specific gene regulatory sites. The association of LSD1/CoREST with GFI1 on chromatin generally confers transcriptional repression and leukemia cell differentiation block (Maiques-Diaz et al, 2022). Therefore, LSD1 has been considered a promising therapeutic target in AML (Harris et al, 2012; Schenk et al, 2012; Fang et al, 2019). Nonetheless, some leukemic cells are resistant to LSD1 inhibition, and additionally, the antileukemic activity of LSD1 inhibitors as a monotherapy is relatively modest (Cai et al, 2020). Therefore, LSD1 inhibitors are used in combination with other molecules in translational and clinical studies (Salamero et al, 2020; Fiskus et al, 2023; Duy et al, 2019).

CDK6 kinase is also a therapeutic target highly expressed in AML independently of the genetic mutations present (Uras et al, 2020; Aleem and Arceci, 2015). CDK6 mRNA transcription is further increased in the presence of FLT3-ITD mutations, or MLL-fusion oncoproteins (Lopez et al, 2016; Uras et al, 2016; Placke et al, 2014). Moreover, CDK6 is a recognized vulnerability in several molecular subtypes of AML, including MLL-fusions (Placke et al, 2014), FLT3-ITD (Lopez et al, 2016; Uras et al, 2016), NUP98-

[1]Aix Marseille University, INSERM, CNRS, Institut Paoli-Calmettes, CRCM-Cancer Research Center of Marseille, Marseille, France. [2]Signaling, Hematopoiesis and Mechanism of Oncogenesis Laboratory, Marseille, France. [3]CRCM's Integrative Bioinformatics platform, Marseille, France. [4]TrGET preclinical facility, CRCM, Marseille, France. [5]Member of Institut Carnot OPALE (Organization for Partnerships in Leukemia), Paris, France. [6]INSERM U1312, BRIC, Université de Bordeaux, Bordeaux, France.
✉E-mail: paulo.de-sepulveda@inserm.fr; desepulvedap@unicancer.fr

fusions (Schmoellerl et al, 2020), RUNX1/ETO (Martinez-Soria et al, 2018) and AML with HOXA9 overexpression (Zhong et al, 2018). In addition to the expected cell cycle arrest related to its canonical function, CDK4/6 inhibition also induces multiple changes in leukemic cells, by interfering with core physiological processes such as transcription, DNA methylation, chromatin organization, and rewiring of metabolic pathways (Klein et al, 2018). Indeed, CDK6 is a component of several transcriptional complexes (Tigan et al, 2016); it also regulates DNA methyltransferases (Heller et al, 2020) and core metabolic enzymes (Wang et al, 2017). Besides these non-canonical functions, CDK6 displays two additional features of interest: it is necessary for leukemic stem cell activation (Scheicher et al, 2015), and promotes anti-cancer immunity (Goel et al, 2017; Deng et al, 2018). However, CDK4/6 inhibition does not eliminate leukemic cells, and the initial clinical trials on leukemia did not yield noteworthy outcomes.

Given the advantages listed above, we hypothesized that CDK6 inhibitors could prove beneficial in AML when paired with a synergistic complementary drug yet to be identified. Hence, the overall objective of our study was to amplify the effects of CDK6 targeting by incorporating a second small-molecule drug to eradicate leukemic cells.

Here, we conducted a screen to identify drugs and protein targets that enhance the response of AML cells to CDK6 inhibitors. We found that the combined inhibition of CDK6 and LSD1 acted in synergy to promote both blast differentiation and the eradication of leukemic progenitors. Mechanistically, the combination induced massive changes in chromatin accessibility, resulting in increased sensitivity to LSD1 inhibition and the transcription of differentiation genes. Importantly, unlike single-molecule treatments, the combination resulted in permanent alterations of the cells. Our data show that the combinatory targeting of these two critical players in AML has great potential to reduce tumor burden, hindering the emergence of variant clones, and ultimately delaying therapy resistance.

# Results

## Identification of a potent combination therapy based on the use of CDK4/6 inhibitors

We set up a functional screen to discover novel therapies based on the use of CDK4/6 inhibitors combined with other molecules of interest in AML (Fig. EV1A). To that effect, we selected a list of 19 drugs either already used in AML or molecules known to impact the differentiation of hematopoietic cells (Fig. EV1B). We used the MV4-11 AML cell line that harbors two independent poor-prognosis mutations, FLT3-ITD and MLL-AF4, in co-culture with HS-5 human stromal cells, in order to mimic a protective environment. We also chose to pre-treat the cells for 24 h with the CDK4/6 inhibitor palbociclib before incubation with the combination of palbociclib and a second molecule. Based on the one two-punch concept, the pre-treatment could sensitize the cells to the second molecule (Wang et al, 2019; Wang and Bernards, 2018). The first read-out of the screen assessed the expression of the myeloid differentiation marker CD11b; the second read-out was the quantification of leukemic progenitors with colony-forming unit capacity (CFU-L), a functional measure for immature progenitors

(Fig. EV1A). Of the 19 molecules tested, the LSD1 inhibitor tranylcypromine (TCP) and all-trans retinoic acid (ATRA) showed a robust increase in CD11b expression when used in combination with palbociclib, compared to the molecules used alone (Fig. EV1B). However, the combination with TCP resulted in a significant reduction of the number of CFU-L, whereas ATRA did not (Fig. EV1C). Thus, we concluded from this screen that the combination of palbociclib and TCP had the potential to induce both cell differentiation of leukemic blasts and target the immature progenitor compartment.

## Combined palbociclib and TCP treatment triggers AML blast differentiation

Unlike the individual drugs, the combination of palbociclib and TCP administered for 4 days strongly induced the expression of early-CD11b and CD14- as well as late myeloid markers -CD86- in MV4-11 cells. (Fig. 1A,B). Similarly, the combination increased the expression of CD86 in MOLM-14, PL-21, THP-1, SKM1, and Mo7e AML cell lines (Fig. 1C–E; Appendix Fig. S1A,B). Following treatment, analyses of cell morphology by flow cytometry and MGG staining confirmed the acquisition of differentiation features, including increased cell size, decreased nucleus/cytoplasm ratio, lighter cytoplasm staining, and an increased number of vacuoles (Fig. 1F; Appendix Fig. S1A,B).

Next, we evaluated the combined treatment on primary AML patient samples (Table EV1). As with the cell lines, the combination significantly increased the expression of CD11b ($N = 17$) and CD86 ($N = 14$) myeloid markers (Fig. 1G), and induced cell morphology changes consistent with differentiation of the primary leukemic cells (Fig. 1H). Interestingly, even though some samples responded well to individual molecules, all patient samples exhibited a greater response to the combination treatment, regardless of their sensitivity to single therapies. We concluded that AML cell lines, as well as primary AML samples, responded significantly better to the combination than to the single molecules. Interestingly, this efficiency did not rely on a specific AML subgroup or mutation status.

## Combined palbociclib and TCP treatment reduces the number of leukemic progenitors

To assess the effects of the combined treatment on immature progenitors, colony-forming unit (CFU) assays were performed. This assay enables the quantification of the small fraction of cells that represents the immature progenitor compartment in primary samples or colony-forming cells (CFC) in cell lines. Interestingly, the CFC compartment of MV4-11 cells was severely impaired by the combination treatment (Fig. 2A). This effect was also observed on other AML cell lines (Fig. 2B,C; Appendix Fig. S1A,B).

The CFU assay is expected to correlate with the engraftment ability of the cells. To provide further support for the ability of the palbociclib and TCP combination to target leukemia-initiating cells, we engrafted in vitro-treated cells into NSG mice, and mouse survival was monitored (Appendix Fig. S1C). ATRA was used as a control as it also induced MV4-11 cell differentiation, but failed to reduce CFUs (Fig. EV1B). Mice engrafted with TCP- or ATRA-treated cells had a median survival equivalent to controls (Appendix Fig. S1D,E). By contrast, mice engrafted with cells

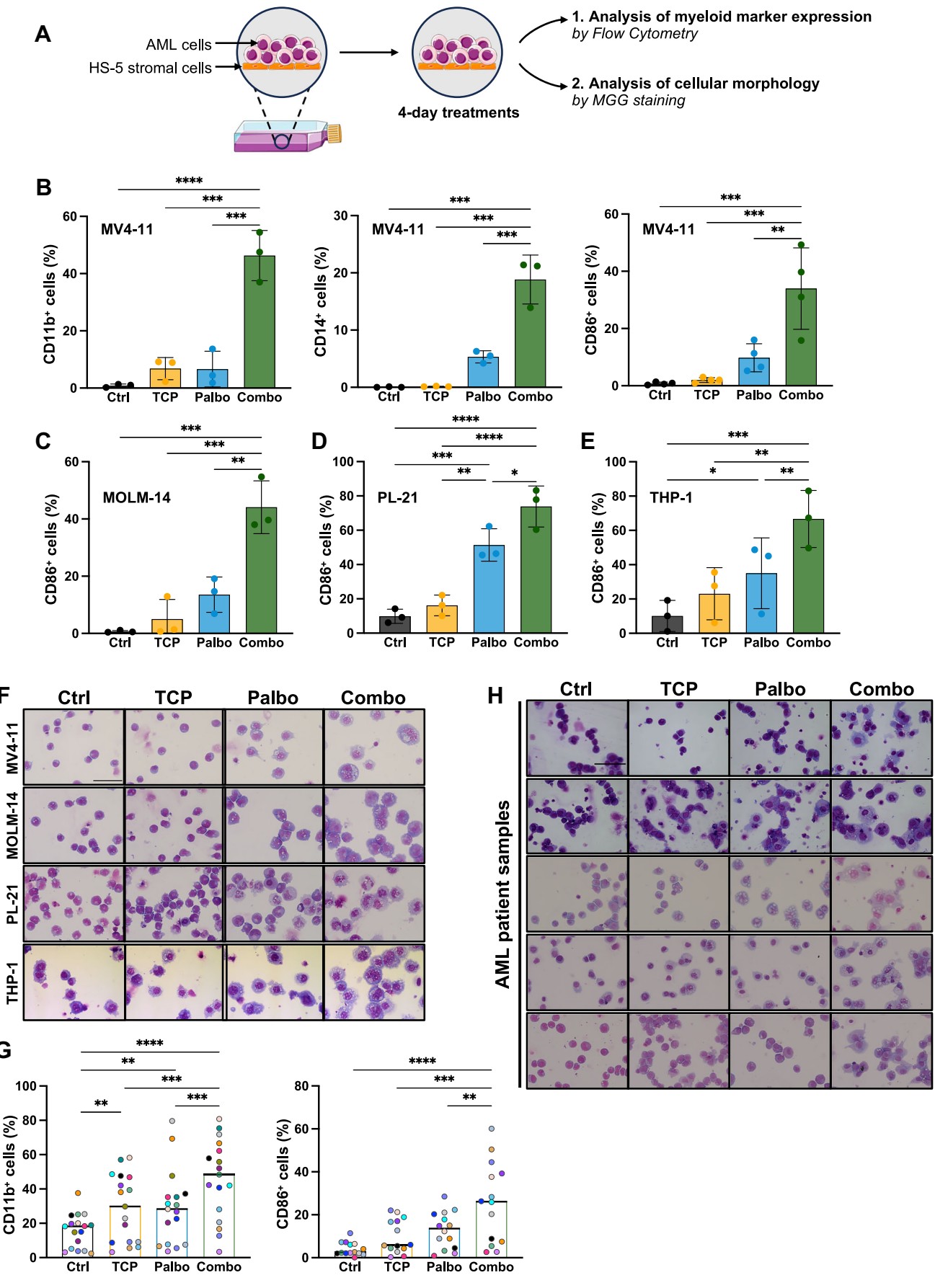

**Figure 1. Palbociclib and TCP cooperate to induce myeloid differentiation in cell lines and AML patient samples.**

(A) Experimental design for the treatment of AML cells. AML cells, cocultured with HS-5 stromal cells, were treated for 4 days with palbociclib at 0.5 μM and/or TCP at 5 μM, and analyzed by flow cytometry and May-Grünwald–Giemsa staining. (B) The percentage of MV4-11 cells expressing CD11b ($n = 3$), CD14 ($n = 3$) or CD86 ($n = 4$) myeloid markers was quantified by flow cytometry following the indicated treatments. Ctrl control, TCP tranylcypromine, Palbo palbociclib, Combo is the combination of palbociclib with tranylcypromine. The exact adjusted $p$ values were as follows: Left panel, Ctrl vs Combo $p < 0.0001$, TCP vs Combo $p = 0.0001$, Palbo vs Combo $p = 0.0001$. Middle panel: Ctrl vs Combo $p = 0.0001$, TCP vs Combo $p = 0.0001$, Palbo vs Combo $p = 0.0008$. Right panel: Ctrl vs Combo $p = 0.0002$, TCP vs Combo $p = 0.0003$, Palbo vs Combo $p = 0.0023$. (C–E) Expression of the late monocytic marker CD86 on MOLM-14 (C) ($n = 3$), PL-21 (D) ($n = 3$), and THP-1 (E) ($n = 3$) AML cell lines. The percentage of cells expressing CD86 was quantified by flow cytometry following the indicated treatments. The exact adjusted $p$ values were as follows: (C) Ctrl vs Combo $p = 0.0003$, TCP vs Combo $p = 0.0005$, Palbo vs Combo $p = 0.0019$. (D) Ctrl vs Combo $p < 0.0001$, TCP vs Combo $p < 0.0001$, Palbo vs Combo $p = 0.0117$, Ctrl vs Palbo $p = 0.0005$, TCP vs Palbo $p = 0.0012$. (E) Ctrl vs Combo $p = 0.0005$, TCP vs Combo $p = 0.0019$, Palbo vs Combo $p = 0.0098$, Ctrl vs Palbo $p = 0.0297$. (F) Cell morphology of the AML cell lines following the treatments. AML cell lines were stained with May-Grünwald–Giemsa. Cells were observed under a Zeiss Apotome microscope. The indicated scale is 50 μm. (G) Primary AML patient cells were treated with the indicated treatments for 96 h. After treatment, cells were analyzed by flow cytometry for the expression of CD11b ($n = 17$) and CD86 ($n = 14$). Each colored dot represents a sample. Exact adjusted pvalues: left panel, Ctrl vs TCP $p = 0.0070$, Ctrl vs Palbo $p = 0.0059$, Ctrl vs Combo $p < 0.0001$, TCP vs Combo $p = 0.0004$, Palbo vs Combo $p = 0.0005$; right panel, Ctrl vs Combo $p < 0.0001$, TCP vs Combo $p = 0.0001$, Palbo vs Combo $p = 0.0022$. (H) May-Grünwald–Giemsa staining of five primary AML patient samples following the indicated treatments. The indicated scale is 50 μm. Histograms indicate the mean of independent experiments ± SD. Statistical analyses were performed using a one-way ANOVA and followed by Tukey's test. *$p < 0.05$, **$p < 0.01$, ***$p < 0.001$, and ****$p < 0.0001$.

treated with the palbociclib and TCP combination showed significant extended survival, indicating that the number of cells capable of initiating leukemia was reduced by the treatment.

We then evaluated these effects on leukemic progenitors in 24 adult and eight pediatric AML primary patient samples (Tables EV1 and EV2), using CFU assays. While individual samples showed very diverse responses to single treatments, all of them were significantly sensitive to the combination (Fig. 2D,E). Indeed, taken individually, all 32 samples invariably showed a reduced number of progenitors after treatment with the combination compared to the control and the single-molecule treatments, highlighting the superiority of the combination therapy on a heterogeneous cohort of samples.

Importantly, purified normal CD34+ cells isolated from human cord blood did not show a significant reduction of progenitors in the CFU assay, suggesting that the in vitro treatment spared normal progenitors (Appendix Fig. S2).

Finally, for six AML primary samples, colonies were collected at the end of the CFU assay, and replated in secondary methylcellulose cultures, without any additional treatment. Interestingly, unlike control and single-molecule treated cells, progenitors initially treated with the combination lost their ability to maintain in culture (Fig. 2F). This indicated that the treatment eradicated leukemic progenitors.

In conclusion, the combination of CDK6 and LSD1 inhibitors targeted the progenitor compartment of AML.

## Combined CDK6 and LSD1 inhibition is synergistic

To exclude that these results were specifically associated with palbociclib and TCP molecules, we used two additional CDK4/6 inhibitors, ribociclib and abemaciclib, and two other LSD1 inhibitors, ORY-1001 and GSK2879552. As illustrated in Fig. EV2A–C, comparable results were obtained using ORY-1001 or ribociclib, on both cell differentiation and reduction of leukemic progenitors. Furthermore, all nine combinations of CDK6 and LSD1 inhibitors acted synergistically (Fig. EV2D).

Finally, we also used RNA interference to demonstrate whether these results could be reproduced independently through reduced expression of CDK6 and LSD1. CD11b expression was indeed increased when LSD1 was targeted by RNA interference in

combination with palbociclib (Fig. EV3A), or when CDK6 expression was reduced in combination with LSD1 inhibitors (Fig. EV3B). Furthermore, the number of CFCs dropped dramatically when both CDK6 and LSD1 expression were decreased through RNA interference, mirroring the results observed with the pharmacological inhibitions (Fig. EV3C).

## Superior effects of the CDK6/LSD1 combination on leukemic progenitors compared to dual targeting of FLT3 and LSD1

A functional regulation between FLT3 and CDK6 had previously been demonstrated. Indeed, both proteins mutually regulate their expression (Uras et al, 2016), and CDK6 is required for FLT3-ITD-mediated cell transformation (Lopez et al, 2016). Furthermore, the benefits of the dual targeting of FLT3 and LSD1 have been reported recently (Yashar et al, 2023). Given these findings, we evaluated whether FLT3 inhibition was equivalent to CDK6 inhibition. To this end, MV4-11 cells (which harbor a FLT3-ITD mutation) were treated with the FLT3 inhibitor AC220 (quizartinib) alone or in combination with TCP, as above. Unlike CDK6 targeting, FLT3 inhibition had a very modest impact on both cell differentiation and the leukemic CFC compartment when combined with TCP (Appendix Fig. S3). We concluded that the combined CDK6/LSD1 targeting induced unique features, different from dual FLT3/LSD1 targeting.

## Inhibition of CDK6 and LSD1 reduces the leukemic burden of preclinical mouse models of AML

To evaluate the combination treatment in vivo, we first used the MOLM-14-Luciferase cell line that allows the monitoring of the leukemic burden by live imaging. After transplantation, this model induced acute leukemia within a few weeks with a strong homing of the cells in the bone marrow, and few cells residing in the spleen. NSG mice transplanted with this cell line were treated with LSD1 inhibitor ORY-1001 or palbociclib alone, or with the ORY-palbociclib combination 4 days per week for 4 weeks. ORY-1001 was selected for in vivo experiments for two reasons: first, we observed less toxicity in mice compared to TCP, and second, it was the most advanced LSD1 inhibitor in AML clinical trials. At day 24 post-transplantation, live-imaging analysis revealed a reduction in

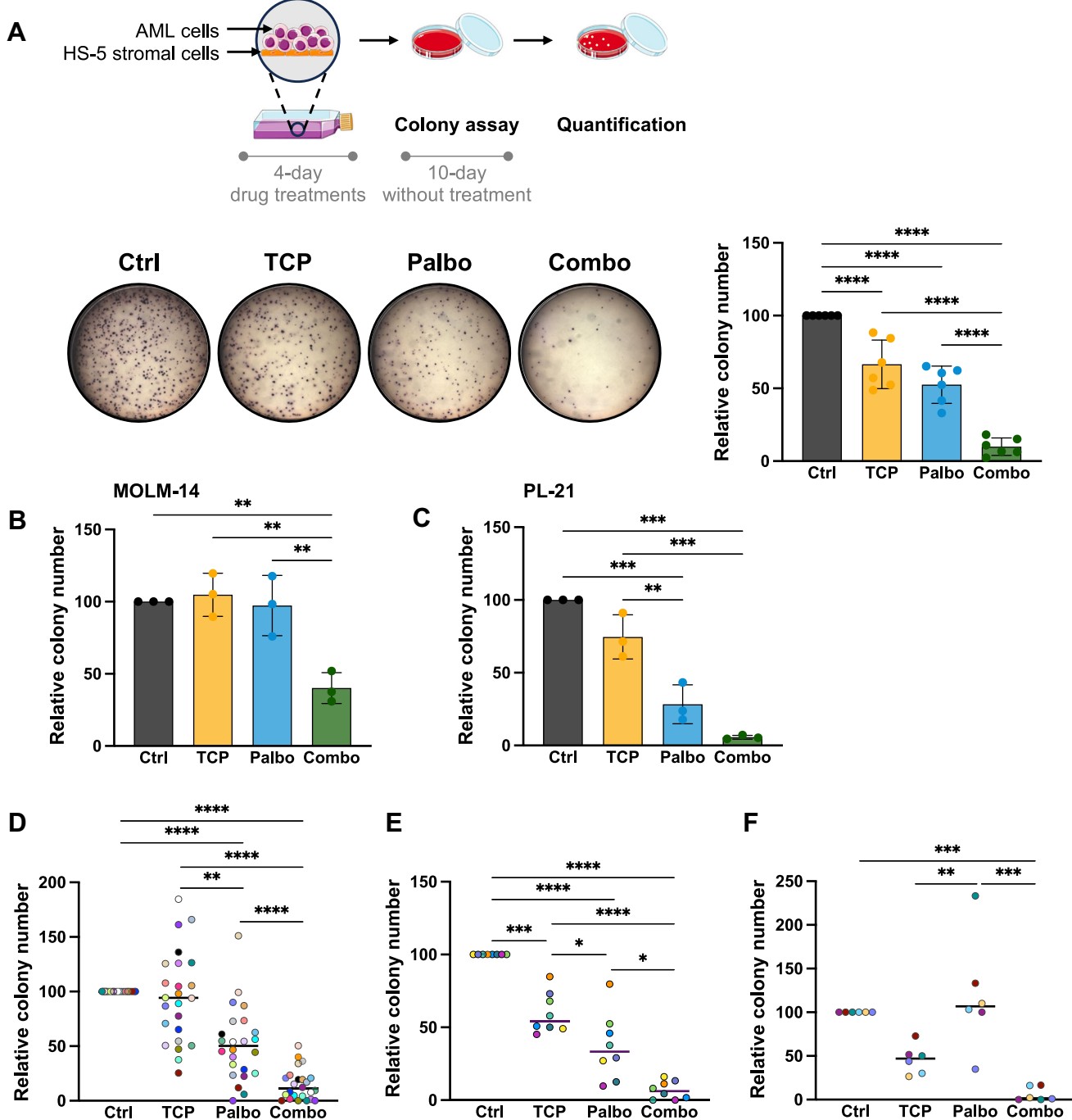

leukemic burden with single agents, but the combination treatment demonstrated superior efficacy (Fig. 3A). These results were confirmed by the analysis of bone marrow and spleens (Fig. 3B,C). Indeed, the combination treatment led to a strong and significant decrease in leukemic cells, especially in the bone marrow, the primary colonized hematopoietic organ in this model.

We additionally administered the combination to wild-type C57BL/6 mice to assess its impact on normal hematopoiesis. After 4 weeks of treatment, blood counts were similar in control and treated mice, except for moderately elevated platelet levels (Fig.

EV4A). In the bone marrow, the number of cells was identical to control mice, and the proportion of myeloid progenitors was unaffected (Fig. EV4A). We concluded that combined inhibition efficiently decreased leukemic burden in vivo, while preserving normal haematopoiesis.

Next, we evaluated the combination on five patient-derived xenograft (PDX) mouse models of AML. These primary AML samples had diverse combinations of mutations (Table EV1). Mice were treated once leukemic blasts were detected in the bloodstream, and the treatments lasted for 3 to 5 weeks, until control mice

**Figure 2. Combination of palbociclib and TCP durably alters colony formation.**

(A) Top panel: Experimental design to quantify colony-forming cells of treated AML cells. MV4-11 cells were treated for 4 days with palbociclib (0.5 μM), TCP (5 μM), or the combination of palbociclib and TCP. Bottom panel: Representative images of methylcellulose colony formation assays and the quantification of the colonies (mean ± SD; $n = 6$), relative to control. Exact adjusted $p$ values: Ctrl vs TCP $p < 0.0001$; Ctrl vs Palbo $p < 0.0001$; Ctrl vs Combo $p < 0.0001$; TCP vs Combo $p < 0.0001$; Palbo vs Combo $p < 0.0001$. (B, C) Quantification of the colonies on MOLM-14 and PL-21 AML cells, treated as described in (A), relative to control. Data represent the mean ± SD of $n = 3$ independent experiments. In (B), adjusted $p$ values: Ctrl vs Combo $p = 0.0044$, TCP vs Combo $p = 0.0030$, Palbo vs Combo $p = 0.0056$. In (C), Ctrl vs Palbo $p = 0.0005$, Ctrl vs Combo $p = 0.0001$, TCP vs Palbo $p = 0.0052$, TCP vs Combo $p = 0.0006$. (D, E) Effects of the indicated treatments on CFCs of primary patient AML cells. (D) Adult samples ($n = 24$); (E) pediatric samples ($n = 8$). Cells treated with the indicated molecules for 96 h were then seeded in methylcellulose and colonies counted after 10 days. The number of colonies is represented as a relative number of colonies compared to control cells. Scatter plots indicate the median. Each colored dot represents a sample. The exact adjusted pvalues were: (D) Ctrl vs Palbo $p < 0.0001$, Ctrl vs Combo $p < 0.0001$, TCP vs Combo $p < 0.0001$, Palbo vs Combo $p < 0.0001$ ; (E) Ctrl vs TCP $p < 0.0001$, Ctrl vs Palbo $p < 0.0001$, Ctrl vs Combo $p < 0.0001$, TCP vs Palbo $p = 0.0035$, TCP vs Combo $p < 0.0001$, Palbo vs Combo $p = 0.0002$. (F) Secondary replating assays from colonies obtained in (D) for six adult patient samples ($n = 6$). Scatter plots indicate the median, with each colored dot representing a sample. The exact adjusted $p$ values: Ctrl vs Combo $p = 0.0009$; TCP vs Palbo $p = 0.0078$; Palbo vs Combo $p = 0.0001$. Statistical analyses were performed using a one-way ANOVA followed by Tukey's test. *$p < 0.05$, **$p < 0.01$, ***$p < 0.001$, and ****$p < 0.0001$.

displayed signs of disability associated with leukemia development. In these models, 4 out of 5 PDX mice treated with the combination of CDK6 and LSD1 inhibitors showed reduction of leukemic cells in both bone marrow, spleen and blood (Figs. 3D and EV4B), with greater efficacy compared to single drug treatments (Fig. 3E,F). Interestingly, the remaining leukemic cells in PDX mice treated with the combination showed increased expression of monocyte late differentiation markers CD86 and CD163, along with reduced CFU activity (Fig. EV4C). Thus, the main observations made in vitro on AML cells were recapitulated in vivo. In conclusion, the combination proved efficient in reducing the leukemic burden in four mouse models of AML.

## Long-lasting effects of dual CDK6 and LSD1 inhibition

A key limitation of CDK6 inhibitors lies in the reversibility of their effects. We thus analysed whether the combination therapy provided long-lasting effects on leukemic cells. First, PL-21 and MOLM-14 cell lines were treated for 4 days, washed and labeled with CFSE to monitor cell doubling. Cells treated with TCP or palbociclib had doubling times comparable to the control, with palbociclib-treated cells presenting a slight delay (Fig. 4A; Appendix Fig. S4). By contrast, cells treated with the combination showed major impaired divisions (Fig. 4A; Appendix Fig. S4).

MV4-11 cells were also serially replated in methylcellulose to monitor the long-term effects of the drugs. While palbociclib-treated cells retained the same capacity to form colonies as the control cells, cells treated with the combination were exhausted after the third plating, demonstrating the long-term effects of the combination (Fig. 4B, middle and right panels).

Using the same starting protocol, we also assessed the effects of successive treatments on the remaining CFCs, after the first and second replating on methylcellulose (Fig. 4C). In these settings, the combination eradicated the cells with CFU potential (Fig. 4C,D). We concluded that the combination of CDK6 and LSD1 inhibitors triggered long-lasting effects on cell cycle arrest and that it could eradicate immature cells with CFU potential.

## Dual CDK6/LSD1 targeting restores the expected transcriptional response associated with LSD1 inhibition

In order to gain insight into the molecular pathways and biological processes that are modified by the combination, we performed RNA-seq analyses on AML cells. Differentially expressed genes (DEG) analysis revealed that TCP had a modest impact on gene expression (96 DEG; log2 fold change $\geq 1$, $p_{adj} < 0.01$), while palbociclib (1487 DEG) and the combination (2016 DEG) induced major changes (Fig. 5A,B; Appendix Fig. S5). Moreover, the number of DEG triggered by the combination was higher than the addition of the single molecules. Accordingly, more pathways were modified by the combination, mainly corresponding to processes related to AML differentiation, cell cycle arrest, metabolism and cell death (Fig. 5E). Consistent with the flow cytometry, cytology and CFU data (Figs. 1 and 2), GSEA analyses confirmed that the combination enhanced the expression of myeloid differentiation genes and the loss of LSC/immature-cell signature (Fig. 5C).

Next, we interrogated the existing signatures associated with LSD1 inhibition. Interestingly, the combination induced a greater response to LSD1 inhibition than TCP alone (Fig. 5D). This was validated by the analysis of seven of the DEGs either by qPCR ($N = 5$) or protein expression using flow cytometry ($N = 3$) (Appendix Fig. S6). LSD1 inhibitors disrupt protein-protein interaction with GFI1 at the chromatin, allowing access to SPI1 and other transcription factors to critical chromosomal regulatory regions (Barth et al, 2019; Maiques-Diaz et al, 2018). To confirm that the combination therapy restores a greater response to LSD1 inhibition, we took advantage of two gene signatures that define SPI1 upregulated and downregulated genes in myeloid cells (McKenzie et al, 2019). Venn diagrams highlighted that more SPI1-related genes were deregulated by the combination compared to the LSD1 inhibitor alone (426 vs 214 genes for the expected upregulated genes and 271 vs 76 genes for the expected downregulated genes) (Appendix Fig. S7A). Furthermore, GSEA analyses confirmed that cells treated with the combination were positively correlated with the expected SPI1 upregulated signature and negatively correlated with the expected SPI1 downregulated signature, indicating that the combination restored the SPI1 transcriptional response (Appendix Fig. S7B). Taken together, these results indicated that the combination enhanced the sensitivity of leukemic cells to LSD1 inhibition.

Finally, RNA-seq analysis was conducted on six primary AML samples (Fig. 6A,B). Despite the anticipated diversity among primary AML patient samples, all six samples taken individually showed a consistent enrichment of the myeloid differentiation signature when treated with the combination (Fig. 6C,F; Appendix Fig. S8A). Remarkably, and similar to what was observed with the MV4-11 AML model, the combination treatment also triggered the loss of the LSC/immature signatures (Fig. 6D) and the restoration

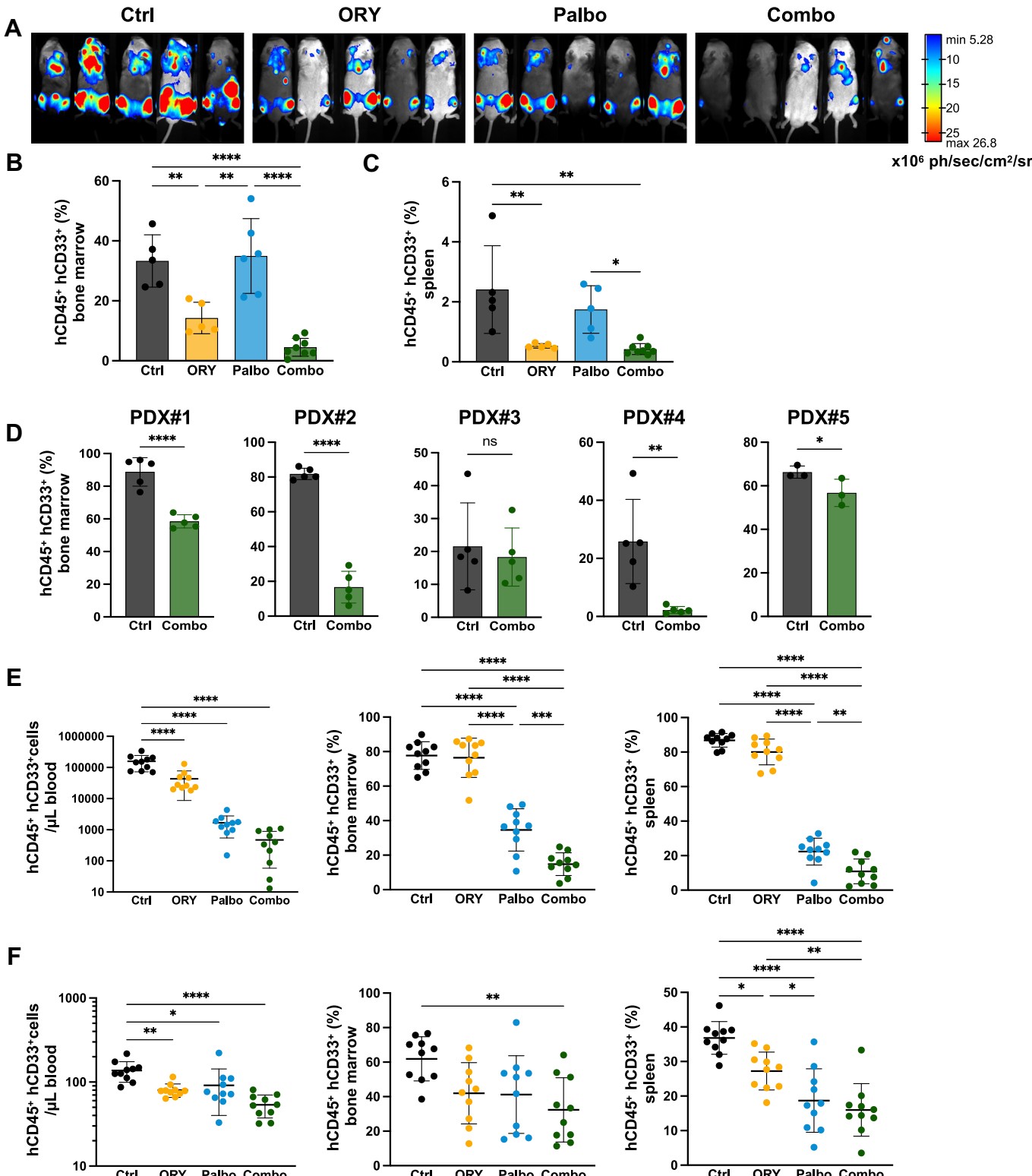

of a LSD1 inhibition response as evidenced by GSEA (Fig. 6E), and heatmap analyses of the individual primary samples (Fig. 6G). Hence, the analyses of primary AML samples supported the previous conclusions drawn from the AML cell line model.

Interestingly, the comparison of the DEGs from both MV4-11 and primary samples revealed a strong overlap, with 523 common deregulated genes (Appendix Fig. S8B).

**Figure 3.   The combination reduces AML progression in vivo.**

(A) Bioluminescent imaging of NSG mice at day 24, post-xenotransplantation of $2 \times 10^5$ luciferase-labeled MOLM-14 cells. Mice were treated by intraperitoneal injections of either palbociclib (25 mg/kg), ORY-1001 (0.0125 mg/kg), the combination of the two molecules or vehicle control, for 4 weeks. (B, C) Quantification of the percentage of MOLM-14 cells defined as hCD45 + hCD33+ cells, by flow cytometry, in the bone marrow (B) and in the spleen (C). Data represent the mean ± SD of one experiment with $n = 5$ mice per condition. The adjusted pvalues were: (B) Ctrl vs ORY $p = 0.0057$, Ctrl vs Combo $p < 0.0001$, ORY vs Palbo $p = 0.0018$, Palbo vs Combo $p < 0.0001$ ; (C) Ctrl vs ORY $p = 0.0055$, Ctrl vs Combo $p = 0.0012$, Palbo vs Combo $p = 0.0337$. (D) Quantification of hCD45 + hCD33+ leukemic blast cells in NSG mice transplanted with primary AML patient cells. Treatment with the combination of palbociclib and ORY-1001 was initiated once the engraftment was confirmed by the appearance of blast cells in the bloodstream, and were administered for 3–6 weeks until an endpoint was reached. The percentage of human myeloid cells was assessed by flow cytometry in the bone marrow. Data represent the mean ± SD. Statistical analysis was performed using an unpaired one-tailed Student's $t$-test. PDX#1 $p < 0.0001$ ; PDX#2 $p < 0.0001$ ; PDX#3 $p = 0.3301$ ; PDX#4 $p = 0.0033$ ; PDX#5 $p = 0.0368$. (E, F) Flow cytometry quantification of hCD45 + hCD33+ leukemic blast cells in blood, bone marrow and spleen of NSG mice engrafted with PDX#2 (E) and PDX#4 (F), treated as indicated. Scatter plots indicate the mean ± SD of an experiment with $n = 10$ mice per condition. Adjusted $p$ values in (E): Left panel, Ctrl vs ORY $p < 0.0001$, Ctrl vs Palbo $p < 0.0001$, Ctrl vs Combo $p < 0.0001$ ; Middle panel, Ctrl vs Palbo $p < 0.0001$, Ctrl vs Combo $p < 0.0001$, ORY vs Palbo $p < 0.0001$, ORY vs Combo $p < 0.0001$, Palbo vs Combo $p = 0.0004$; Right panel, Ctrl vs Palbo $p < 0.0001$, Ctrl vs Combo $p < 0.0001$, ORY vs Palbo $p < 0.0001$, ORY vs Combo $p < 0.0001$, Palbo vs Combo $p = 0.0030$. (F) Left panel, Ctrl vs ORY $p = 0.0033$, Ctrl vs Palbo $p = 0.0230$, Ctrl vs Combo $p < 0.0001$; Middle panel, Ctrl vs Combo $p = 0.0047$; Right panel, Ctrl vs ORY $p = 0.0199$, Ctrl vs Palbo $p < 0.0001$, Ctrl vs Combo $p < 0.0001$, ORY vs Palbo $p = 0.0449$, ORY vs Combo $p = 0.0049$. Except for (D), statistical analyses were performed using a one-way ANOVA followed by Tukey's test. *$p < 0.05$, **$p < 0.01$, ***$p < 0.001$, ****$p < 0.0001$, and ns for not significant.

## Dual CDK6/LSD1 targeting remodeled chromatin accessibility

In parallel, chromatin accessibility was also assessed by ATAC-seq. Interestingly, chromatin accessibility was largely unchanged upon LSD1 inhibition by TCP (Fig. EV5A; Appendix Fig. S9A,B). By contrast, CDK6 inhibition resulted in 335 accessibility changes (with 158 losses and 177 gains). Notably, the combination of TCP and palbociclib further increased chromatin remodeling, with 453 losses and 490 gains in accessibility ($p_{adj} < 0.05$). The combination modulated the gain of accessibility in chromatin regions regulating genes associated with cell differentiation (Fig. EV5B), such as MAFB, SPI1, LYS, RUNX2, ITGA4, LILRB1, and S100A9, and the loss of accessibility of regulatory regions of genes involved in the immature state (LMO2, MYC). Both palbociclib alone and the combination mainly affected intronic (~47%) and intergenic regions (~42%), as well as transcription start sites (~10%) (Appendix Fig. S9C). Importantly, DNA sites showing gain of accessibility were enriched for motifs predicted to recruit two main transcription factors involved in myeloid differentiation, CEBPA and SPI1 (Fig. EV5C). Interestingly, independent analyses using the TOBIAS genomic framework confirmed that both CEBPs and SPI1 are among the transcription factors exhibiting the greatest increase in footprints within the ATAC-seq dataset (Fig. EV5D). This suggested that occupancy of both CEBPs and SPI1 at genomic regulatory regions are increased following the treatment.

Remarkably, the expression of DEGs strongly correlated with chromatin accessibility at their assigned regulatory regions, indicating that changes in chromatin accessibility is a key mechanism responsible for the observed gene expression changes (Fig. EV5E). This correlation is particularly driven by the genes upregulated by the combination (listed in Table EV3). We concluded that the combination therapy remodeled chromatin accessibility, unlocking regulatory regions critical for promoting myeloid differentiation.

## Dual CDK6/LSD1 inhibition induces cell death by ferroptosis

The RNA-seq analysis unveiled that several genes deregulated by the combination treatment were involved in cell death pathways, particularly genes related to apoptosis (Fig. 5E; Appendix Fig. S10A). GSEA analyses confirmed that apoptosis and ferroptosis gene signatures were enriched (Appendix Fig. S10B,C), prompting us to investigate further cell death following treatment. Annexin V staining as well as the quantification of cells with sub-G1 DNA content confirmed that the combination of palbociclib and TCP increased cell death (Fig. 7A,B). Interestingly, cell death persisted after removal of the drugs (Fig. 7A, right panel), indicating that the treatment generated irreversible damage to the cells.

We then used small-molecule inhibitors to elucidate the cell death pathway preferentially involved. As illustrated in Fig. 7C,D, only Ferrostatin-1, but no other inhibitor, protected against treatment-induced cell death. Additionally, the lipid peroxidation sensor C11-BODIPY 581/591 confirmed that the combined treatment led to the accumulation of oxidized lipids (Fig. 7E,F). Finally, analyses of primary AML samples revealed both cell death, and oxidation of lipids following incubation of the cells with the combination (Appendix Fig. S10D,E). We concluded that, in addition to the cell cycle arrest and differentiation effects, the combination induced ferroptosis-dependent cell death.

## Discussion

In this study, we report that the combined inhibition of CDK6 and LSD1 induces unique features in AML cells. This combination is a potent blast differentiation therapy, and targets the immature progenitor population with self-renewal capacity. Importantly, it is efficient on all tested AML cell lines and primary samples, including cells resistant to cytarabine (four of the cell lines used here), FLT3 inhibitors (4 PDX samples used in the study) and primary samples at relapse. As such, this therapy fulfils the need to target the majority of leukemic cells, as well as cells with repopulating potential. Remarkably, the combination was potent in all samples, and it consistently demonstrated greater effectiveness compared to individual molecules in primary AML samples, even among those that were highly sensitive to a single agent. In addition, the combination offers several benefits, such as synergy, lasting effects even after drug removal, and induction of cell death.

We investigated the impact of combined CDK6 and LSD1 inhibition on progenitor cells from primary AML samples using colony assays. The observed reduction in colony formation is likely attributed to the targeting of leukemic progenitors, supported by the high blast

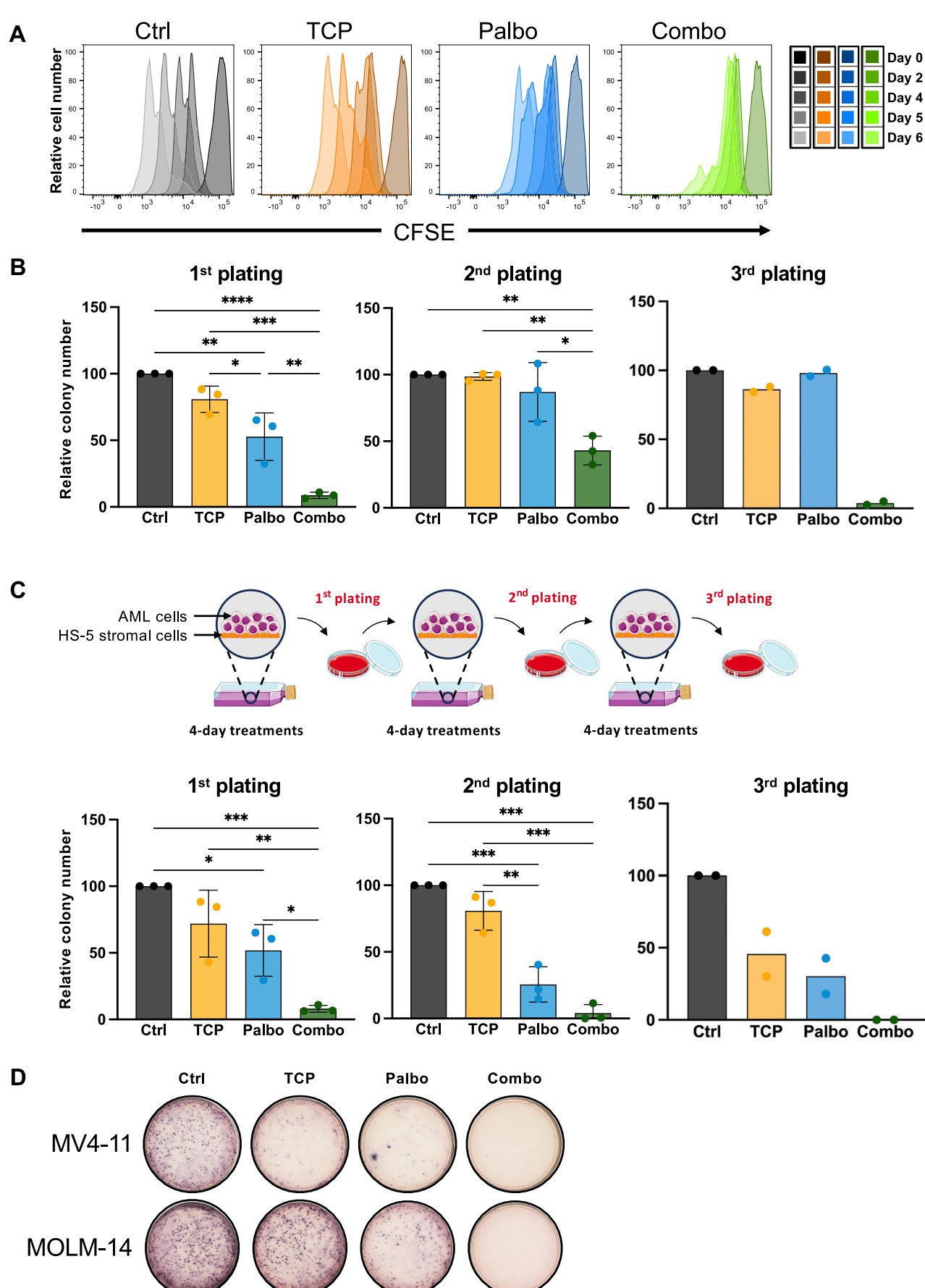

**Figure 4. Lasting effects of combined treatment on cell proliferation and colony-forming capacity of AML cells.**

(A) CFSE cell proliferation analysis in PL-21 AML cell line. Cells treated as described before for 96 h, were labeled with CFSE to monitor cell division over the following 6 days. (B) Serial plating assays of MV4-11 on methylcellulose following a single round of treatment. MV4-11 cells were treated as previously. After the 4-day treatments, an equal number of cells was seeded in methylcellulose for 10 days ($n = 3$), and two serial replatings were performed (second and third plating; $n = 3$ and $n = 2$, respectively) without any additional treatment. Left panel, Ctrl vs Palbo $p = 0.0024$, Ctrl vs Combo $p < 0.0001$, TCP vs Palbo $p = 0.0303$, TCP vs Combo $p = 0.0002$, Palbo vs Combo $p = 0.0035$. Middle panel, Ctrl vs Combo $p = 0.0028$, TCP vs Combo $p = 0.0032$, Palbo vs Combo $p = 0.0104$. (C) Serial treatments and replating assays of MV4-11 cells. Top panel: Experimental design for the treatments and replating strategy. After the 4-day treatment cycle, MV4-11 cells were seeded into methylcellulose for 10 days. Bottom panel: quantification of the remaining colonies over three rounds of replating. First plating, $n = 3$; second plating, $n = 3$; third plating, $n = 2$. Left panel, Ctrl vs Palbo $p = 0.0147$, Ctrl vs Combo $p = 0.0005$, TCP vs Combo $p = 0.0037$, Palbo vs Combo $p = 0.0229$. Middle panel, Ctrl vs Palbo $p = 0.0005$, Ctrl vs Combo $p = 0.0001$, TCP vs Palbo $p = 0.0024$, TCP vs Combo $p = 0.0004$. (D) Representative pictures of colonies obtained in the third round of replating of MV4-11 cells (top panel) and MOLM-14 cells (bottom panel). Statistical analyses are performed using a one-way ANOVA test followed by Tukey's test. $*p < 0.05$, $**p < 0.01$, $***p < 0.001$, and $****p < 0.0001$.

percentage in the patient samples and the lack of inhibitory effect on normal CD34$^+$ hematopoietic cells. However, the leukemic origin of the colonies was not directly confirmed through analysis of the colonies themselves. CDK6 is a critical vulnerability of AML cells, not compensated by CDK4 or other CDK proteins (Lopez et al, 2016). CDK6 dependence appears to be a common feature of all AML cells, regardless of their mutation status, making it a recognized target of interest in AML. Beyond inducing cell cycle arrest, CDK6 inhibition has several other consequences in leukemia: it promotes the expression of cell differentiation markers, and, in some models, induces cell death (Uras et al, 2016). However, these effects are modest and reversible. Therefore, CDK6 inhibition was investigated in combination with either chemotherapy or other small-molecule inhibitors. Similarly, LSD1, due to its role in maintaining the chromatin landscape that shapes the identity of immature cells, has tremendous potential as a therapeutic target in cancer. In AML, the association of LSD1 inhibition with either ATRA (Schenk et al, 2012; Wass et al, 2021) or azacytidine (phase II trial of iadademstat in combination with azacytidine, EudraCT No.: 2018-000482-36) yielded promising results, but these did not lead to conclusive clinical trials. Therefore, the best combination of molecules has yet to be determined. We demonstrate here that the combination of CDK6 and LSD1 inhibitors is synergistic in inducing differentiation of leukemic blasts and reducing the number of immature leukemic progenitors. Although both CDK4/6 and LSD1 inhibitors have been described as promoting differentiation, their efficacy is marginal in most AML cells. We thus argue that the combination efficiently unlocks the differentiation block in AML cells, through their complementary roles in modifying chromatin accessibility and transcription.

Mechanistically, our molecular analyses revealed that the drug combination increased chromatin accessibility at critical transcription factor regulatory sites, which include the myeloid differentiation factors CEBP and SPI1. Importantly, changes in chromatin accessibility correlated with deregulated gene expression, strongly suggesting that the combination partially restored the myeloid differentiation program by acting at the chromatin level.

Several AML models used in the study carried MLL-rearrangements, including PDX#2, which showed the strongest response in vivo. Given the poor prognosis associated with this category, further preclinical evaluation in MLL-rearranged AML is warranted.

In addition to the expected effect on cell cycle arrest and myeloid differentiation, the combined targeting of CDK6 and LSD1 also induced cell death. Previous studies have reported that either CDK6 or LSD1 single inhibition can lead to cell death (Uras et al, 2016; Leiendecker et al, 2020). We show here that cell death is a major response to the combination in vitro, and our results suggest that it

occurs through ferroptosis. Although cell death was not seen in vivo in our mouse models, it should be noted that the treatment was discontinued due to the extreme sensitivity of the transplanted NSG mouse models. Furthermore, the combination triggered durable therapeutic responses, even after removal of the drugs. This sustained effect was not observed with the single molecules, representing a significant improvement for therapies based on CDK4/6 inhibitors such as palbociclib. This feature holds considerable promise for accomplishing the objective of eliminating leukemic cells.

LSD1 inhibitor ORY-1001 (iadademstat) and FLT3 inhibitor were combined in a clinical trial for patients with mutated-FLT3 relapsed/refractory AML (Clinical trial NCT05546580). This combination specifically targets the FLT3 oncogenic mutation, but not the minor clones lacking this mutation. Importantly, our data indicate that it failed to reduce the number of mutant FLT3 leukemic progenitors in the CFU assay. Similarly, iadademstat was evaluated in combination with azacytidine (Salamero et al, 2020) and in the phase IIa Alice Trial, showing significant efficacy in previously untreated, unfit AML patients (Salamero et al, 2022). As azacytidine, CDK6 inhibition also reduces the activity of DNA methyltransferases (Heller et al, 2020). Therefore, there may be overlapping activities in both combinations. A preclinical study comparing the two combinations may be of interest to quantify and highlight the advantages of each of these dual combinations of molecules.

Resistance to therapy is a major challenge in AML. One major advantage of CDK6 inhibition is that cells resistant to cytarabine or FLT3 inhibitors remain sensitive (our unpublished results). LSD1 inhibitors impact the epigenetic landscape associated with drug resistance. Indeed, LSD1 inhibition has the potential to overcome epigenetic resistance by facilitating the binding of SPI1/PU.1 and IRF8, thereby regulating the expression of differentiation genes (Bell et al, 2019). Combining drugs with distinct mechanisms of action and demonstrating synergistic effects can help overcome or delay the development of resistance. CDK4/6 inhibitors and LSD1 inhibitors may have complementary effects in bypassing resistance mechanisms that arise with single-agent therapy. This combination could lead to more profound and sustained inhibition of leukemic cell proliferation.

Understanding how cancer cells suppress the immune response is crucial to anticipate future therapies. It has been reported previously that both CDK6 and LSD1 single inhibition increases the immune response against leukemic cells (Klein et al, 2018; Goel et al, 2017; Deng et al, 2018; Sheng et al, 2018; Mamun et al, 2023). Therefore, elucidating the mechanisms by which CDK6 and LSD1 function to suppress the immune response, and how their respective inhibitions unleash the anti-cancer immune response, is a challenge that may pave the way for the development of novel immunotherapy approaches.

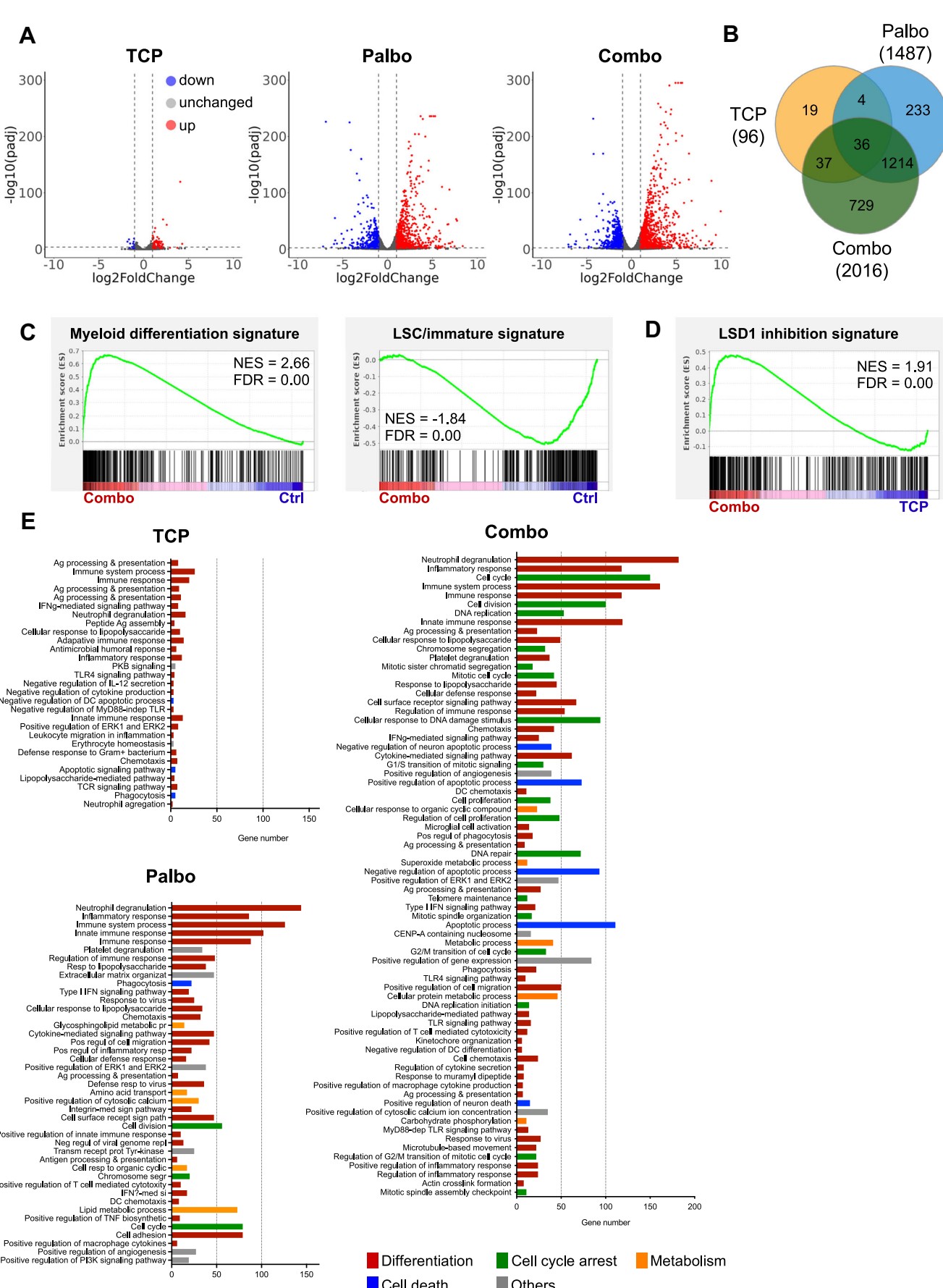

◀ **Figure 5. Gene expression analysis reveal enhanced LSD1 inhibition and increased chromatin accessibility at CEBP and SPI1 sites.**

RNA-seq analyses of MV4-11 cells treated for 4 days with the combination or the single molecules as indicated. (A) Volcano plots highlighting the differentially expressed genes (DEG) following treatments ($n = 2$) with either TCP, palbociclib or the combination, compared to control cells ($p_{adj} < 0.01$). Statistical significance was assessed using the Wald test, and the resulting $p$ values were adjusted for multiple testing using the Benjamini–Hochberg procedure to control the false discovery rate. (B) Venn diagram indicating the number of DEG compared to the control (log2 fold change > 1, $p_{adj} < 0.01$). (C) Gene set enrichment analyses (GSEA) of myeloid differentiation and LSC signature, comparing the combination treatment to the control. Statistical analyses were performed using the GSEA software. (D) GSEA of the LSD1 inhibition signature, comparing the cells treated with the combination with TCP only. Statistical analyses were performed using the GSEA software. (E) Biological processes associated with the deregulated genes, using GeneAnalytics.

# Methods

### Reagents and tools table

| Reagent/resource | Reference or source | Identifier or catalog number |
| --- | --- | --- |
| **Experimental models** | | |
| MV4-11 cells (*H. sapiens*) | Cell Biology Institute (Okayama, Japan) | N/A |
| MOLM-14 cells (*H. sapiens*) | Cell Biology Institute (Okayama, Japan) | N/A |
| PL-21 cells (*H. sapiens*) | DSMZ | ACC 536 |
| SKM1 cells (*H. sapiens*) | DSMZ | ACC 547 |
| THP-1 cells (*H. sapiens*) | ATCC | TIB-202 |
| MOLM-14-*luciferase* cells (*H. sapiens*) | doi: 10.1158/0008-5472.CAN-22-3682. | |
| HS-5 cells (*H. sapiens*) | ATCC | CRL-3611 |
| Primary patient samples | Institut Paoli-Calmettes (Marseille), HIMIP (Hematologie INSERM Midi-Pyrenees) | Table EV1, Table EV2 |
| Purified CD34+ cells | Etablissement Français du Sang (EFS, Marseille) | N/A |
| NOD-SCID-common-γchain-knockout (NSG) mice (*M. musculus*) | Charles River | N/A |
| C57BL/6 J mice (*M. musculus*) | Charles River | N/A |
| **Antibodies** | | |
| ERK2 | Santa Cruz | sc-1647 |
| LSD1 | Cell Signaling Technology | 2139 |
| CDK6 | Cell Signaling Technology | 3136 |
| HRP-conjugated anti-mouse | Cytiva | NA931 |
| HRP-conjugated anti-rabbit | Cytiva | NA934 |
| Lineage cell depletion kit | Miltenyi Biotec | 130-090-858 |
| PE-CF594-streptavidin | BD Biosciences | 562284 |
| CD34-VioGreen | Miltenyi Biotec | 130-124-460 |
| Sca-1-PerCP-Cy5.5 | Biolegend | 122524 |
| c-Kit-APC-eFluor780 | Invitrogen | 47-1171-82 |
| CD16/CD32-PE-Cy7 | Invitrogen | 25-0161-82 |
| CD45-AF700 | Biolegend | 103128 |
| Sytox Blue | Thermo Fisher Scientific | S34857 |

| Reagent/resource | Reference or source | Identifier or catalog number |
| --- | --- | --- |
| LIVE/DEAD stain kit | Thermo Fisher Scientific | L10120 |
| Human CD45-Pacific Blue | Biolegend | 304029 |
| Mouse CD45-APC-eFluor 780 | Invitrogen | 47-0451-82 |
| Human CD11b-PE | Beckman Coulter | IM2581U |
| Human CD14-BV510 | Biolegend | 367124 |
| Human CD86-V450 | Biosciences | 560357 |
| Human CD163-FITC | BD Pharmingen | 563697 |
| DAPI | Sigma-Aldrich | D9542 |
| CellTrace CFSE | Thermo Fisher Scientific | C34554 |
| PE Annexin V apoptosis detection kit | BD Pharmingen | 559763 |
| BODIPY 581/591 C11 | Thermo Fisher Scientific | D3861 |
| Human CD3 microbeads | Miltenyi Biotec | 130-050-101 |
| Human CD31-BV421 | Biolegend | 303124 |
| Mouse/Human CD82 Coralite- 488 | Proteintech | CL488-66803 |
| Human CD184 (CXCR4)-BV510 | Biolegend | 306536 |
| **Oligonucleotides and other sequence-based reagents** | | |
| siRNA | This study | Table EV4 |
| **Chemicals, enzymes and other reagents** | | |
| RPMI 1640 medium | Thermo Fisher Scientific | 11875093 |
| IMDM medium | Thermo Fisher Scientific | 12440053 |
| Fetal bovine serum | Eurobio | CVFSVF00-01 |
| Penicillin-Streptomycin | Thermo Fisher Scientific | 15140122 |
| L-glutamine | Thermo Fisher Scientific | A2916801 |
| DPBS | Thermo Fisher Scientific | 14190144 |
| Human SCF | Peprotech | 300-07 |
| Human FLT3-L | Peprotech | 300-19 |
| Human IL-3 | Peprotech | 200-03 |
| Human GM-CSF | Berlex | Leukine |
| MethoCult H4230 | StemCell Technologies | 04230 |
| MethoCult H4535 | StemCell Technologies | 04535 |
| Dimethylsulfoxide, DMSO | Sigma-Aldrich | D4540 |
| Calcitriol | Sigma-Aldrich | D1530 |
| 8-CPT | Sigma-Aldrich | C3912 |

| Reagent/resource | Reference or source | Identifier or catalog number |
|---|---|---|
| Quizartinib | Selleckchem | S1526 |
| Retinoic acid | Sigma-Aldrich | R2625 |
| AGI-5198 | Selleckchem | S7185 |
| AGI-6780 | Selleckchem | S7241 |
| Arsenic trioxide | Sigma-Aldrich | 17971 |
| Azacitidine | Selleckchem | S1782 |
| Decitabine | Selleckchem | S1200 |
| Entinostat | Selleckchem | S1053 |
| OTX015 | Selleckchem | S7360 |
| Panobinostat | Selleckchem | S1030 |
| Prima-1Met | Sigma-Aldrich | SML1789 |
| Ruxolitinib | Selleckchem | S1378 |
| SGI-110 | AdooQ bioscience | A12744 |
| Trans-2-phenylcyclopropylamine-HCl (TCP) | Sigma-Aldrich | P8511 |
| Valproic acid | Sigma-Aldrich | V0033000 |
| Venetoclax | Selleckchem | S8048 |
| Vorinostat | Selleckchem | S1047 |
| Palbociclib (in vitro studies) | Biosynth Carbosynth | FA65120 |
| Palbociclib monohydrochloride (in vivo studies) | MedChemExpress | HY-50767A |
| Ribociclib HCl | Biosynth Carbosynth | FR106415 |
| Abemaciclib | MedChemExpress | HY-16297A |
| Iadademstat dihydrochloride (ORY-1001) | MedChemExpress | HY-12782T |
| GSK2879552 2HCl | Selleckchem | S7796 |
| Ferrostatin-1 | MedChemExpress | HY-100579 |
| Necrostatin-1 | MedChemExpress | HY-15760 |
| Chloroquine | MedChemExpress | HY-17589A |
| Q-VD-OPh | MedChemExpress | HY-12305 |
| Hepes | Sigma-Aldrich | H3375 |
| NaCl | Carlo Erba | 479687 |
| Triton X100 | Sigma-Aldrich | X100 |
| Glycerol | Sigma-Aldrich | G5516 |
| EGTA | Sigma-Aldrich | E3889 |
| $MgCl_2$ | Sigma-Aldrich | M8266 |
| Complete mini, EDTA-free (protease inhibitor cocktail) | Roche | 11 836 170 001 |
| Sodium orthovanadate | Sigma-Aldrich | S6508 |
| Bio-Rad Protein Assay Dye Reagent Concentrate (Bradford reagent) | Bio-Rad Laboratories | 5000006 |
| PVDF membrane | Cytiva | 10600023 |
| Bovine serum albumin | Euromedex | 04-100-812-E |

| Reagent/resource | Reference or source | Identifier or catalog number |
|---|---|---|
| ECL prime | Cytiva | RPN2236 |
| MTT | Sigma-Aldrich | M5655 |
| Ammonium chloride solution | StemCell Technologies | 07850 |
| BD Cytofix/Cytoperm™ Fixation/Permeabilization Kit | BD Biosciences | 554714 |
| autoMACS® Running Buffer -MACS separation buffer | Miltenyi Biotec | 130-091-221 |
| RNeasy mini kit | Qiagen | 74104 |
| Stranded mRNA prep, ligation | Illumina | 20040532 |
| ATAC-Seq kit | Active Motif | 53150 |
| **Software** | | |
| ImageJ | Java 8 | N/A |
| FlowJo | BD | v10 |
| SynergyFinder 3.0 | *synergyfinder.fimm.fi* | N/A |
| FastQC | http://www.bioinformatics.babraham.ac.uk/projects/fastqc/ | 0.11.9 |
| Deeptools suite | R/Bioconductor | v3.2.1 |
| hg19 UCSC genome | Bowtie2 https://doi.org/10.1093/nar/gkt214. | v1.6.4 – v2.3.4.3 |
| FeatureCounts | R/Bioconductor https://doi.org/10.1093/bioinformatics/btt656. | N/A |
| DESeq2 | R/Bioconductor https://doi.org/10.1186/s13059-014-0550-8. | v.1.26.0 |
| Clusterprofiler | Gene Ontology (GO) | v4.6.0 |
| GSEA | Java platform https://doi.org/10.1073/pnas.0506580102. | v4.0.3 |
| MarkDuplicates | R/Bioconductor | v2.20.2 |
| MACS2 | R/Bioconductor https://doi.org/10.1186/gb-2008-9-9-r137. | N/A |
| bamCoverage, multiBigwigSummary | R/Bioconductor, deepTools suite https://doi.org/10.1093/nar/gkw257. | v3.2.1 |
| HOMER mergePeaks | R/Bioconductor https://doi.org/10.1093/nar/gkw257. | v4.11 |
| computeMatrix | R/Bioconductor, deepTools suite | v3.5.1 |
| Prism | GraphPad | v5 ; v10 |
| **Other** | | |
| Gene pulser cuvette | Bio-Rad Laboratories | 1652088 |
| Gene Pulser MXcell™ Electroporation System | Bio-Rad Laboratories | 165-2670 |
| SuperFrost microscopic slides | Fisher Scientific | 12302108 |
| Binocular magnifier | Leica MS5 | N/A |
| AutoMACS pro separator | Miltenyi Biotec | N/A |
| ImageQuant 800 | Cytiva | N/A |
| Apotome microscope | Zeiss | N/A |

| Reagent/resource | Reference or source | Identifier or catalog number |
|---|---|---|
| LSR II cytometer | BD | N/A |
| Fortessa cytometer | BD | N/A |
| Irradiator | Rad Source Technologies RS2000 | N/A |
| Hematology analyzer | Balio Diagnostics | OV-360 |
| Illumina NextSeq 500 | Illumina (TGML, Marseille ; iGenSeq, Paris) | N/A |

## Cell lines and reagents

The human AML cell lines MV4-11 and MOLM-14 were obtained from the Cell Biology Institute (Okayama, Japan). The human AML cell lines PL-21 (ACC 536) and SKM1 (ACC 547) were obtained from DSMZ, and the human AML cell line THP-1 (TIB-202) was purchased from ATCC. These cell lines were cocultured on the human stromal cell line HS-5, also obtained from ATCC. All cell lines were maintained in RPMI 1640 medium (Thermo Fisher Scientific) supplemented with 10% heat-inactivated fetal bovine serum (FBS; Eurobio) and 2 mM L-glutamine (Thermo Fisher Scientific) at 37 °C in a humidified incubator containing 5% $CO_2$. Cell lines were obtained from established cell line repositories and utilized at early passages, without de novo authentication in this study. All cell lines have been tested negative for mycoplasma contamination, by PCR. All frozen batches of cells were tested for Mycoplasma contamination.

For drug screening, 19 molecules were tested at their IC30 for their impact on the viability of the MV4-11 cell line, both alone and in combination with palbociclib: calcitriol (1 nM, Sigma-Aldricht, D1530), 8-CPT (50 μM, C3912, Sigma-Aldricht), quizartinib (1 nM, S1526, Euromedex), retinoic acid (5 μM, R2625, Sigma-Aldricht), AGI-5198 (10 nM, S7185, Selleckchem), AGI-6780 (5 μM, S7241, Selleckchem), arsenic trioxide (1 μM, 17971, Sigma-Aldricht), azacitidine (4 μM, S1782, Selleckchem), decitabine (5 μM, S1200, Selleckchem), entinostat (50 nM, S1053, Selleckchem), OTX015 (50 nM, S7360, Selleckchem), panobinostat (1 nM, S1030, Selleckchem), prima-1Met (2 μM, SML1789, Sigma-Aldricht), ruxolitinib (5 μM, S1378, Selleckchem), SGI-110 (5 μM, A12744, AdooQ bioscience), tranylcypromine (5 μM, P8511, Sigma-Aldricht), valproic acid (0.5 mM, V0033000, Sigma-Aldricht), venetoclax (1 nM, S8048, Selleckchem), and vorinostat (50 nM, S1047, Selleckchem). DMSO was used as a vehicle control.

For in vitro studies, CDK6 inhibitors (CDK6i) palbociclib, ribociclib and abemaciclib were purchased from Biosynth Carbosynth (FA65120), Biosynth Carbosynth (FR106415), and MedChemExpress (HY-16297A), respectively. LSD1 inhibitors (LSD1i) trans-2-phenylcyclopropylamine-HCl (TCP), iadademstat dihydrochloride (ORY-1001), and GSK2879552 2HCl were purchased from Sigma-Aldrich (P8511), MedChemExpress (HY-12782T), and Selleckchem (S7796), respectively. Death inhibitors ferrostatin-1 (HY-100579), necrostatin-1 (HY-15760), chloroquine (HY-17589A), and Q-VD-Oph (HY-12305) were obtained from MedChemExpress. All drugs were dissolved in DMSO (Sigma-Aldrich), and the final concentration was not exceeding 0.1%. All in vitro treatments were performed as 1-day CDK6i alone, followed by 3 days with the combination of CDK6i and LSD1i.

For in vivo studies, palbociclib-HCl (HY-50767A) and ORY-1001 (iadademstat dihydrochloride) (HY-12782T), purchased from MedChemExpress, were dissolved in PBS.

For siRNA transfections, a total of 0.8 nmol siRNAs were mixed with $10^7$ cells in 0.5 mL RPMI 1640 medium in 4 mm gap electroporation cuvettes. Electroporations were carried out at room temperature using a Gene Pulser Electroporator II (Bio-Rad Laboratories) at 300 V and 950 μF, as previously described (Voisset et al, 2010). The siRNA sequences used in the study are listed in Table EV4.

## AML patient samples

Short-term ex vivo AML cultures. Primary patient mononuclear cells were isolated using Ficoll, and stored as frozen samples by the tumor bank facilities of Institut Paoli Calmettes/CRCM (Marseille, France) and from HIMIP (Hématologie INSERM Midi-Pyrénées), using standardized procedures. Bone marrow or peripheral blood cells were thawed and cultured in IMDM medium (Thermo Fisher Scientific), supplemented with 20% heat-inactivated FBS (Eurobio), 2 mM L-glutamine (Thermo Fisher Scientific), 100 U/mL Penicillin-Streptomycin (Thermo Fisher Scientific), 100 ng/mL human SCF (300-07, Peprotech), 100 ng/mL human FLT3-L (300-19, Peprotech), 5 ng/mL human IL-3 (200-03, Peprotech), and 10 ng/mL human GM-CSF (Leukine, Berlex) at 37 °C in a humidified incubator containing 5% $CO_2$, at a concentration of $0.5-1 \times 10^6$ cells/mL. All in vitro treatments with the combination of molecules were performed as 1-day CDK6i alone, followed by 3 days with the combination of CDK6i and LSD1i.

The project was approved by the IPC and HIMIP Institutional Review Boards. Marseille tumor bank operates under authorization # AC-2013-1905 granted by the French Ministry of Research. HIMIP collections has been declared to the Ministry of Higher Education and Research (DC 2008-307 collection 1) and obtained a transfer agreement (AC 2008-129) after approbation by the ethical committee "Comité de Protection des Personnes Sud-Ouest et Outremer II". Human purified CD34+ cells were provided by the Etablissement Français du Sang (EFS, Marseille, France), under the agreement number 776-7551. Informed consent was obtained from all subjects. The experiments reported here conformed to the principles set out in the WMA Declaration of Helsinki and the Department of Health and Human Services Belmont Report.

## Western immunoblotting

Cell lysates were prepared at 4 °C in HNTG buffer (50 mM Hepes pH 7.0, 150 mM NaCl, 1% Triton, 10% glycerol, 1 mM EGTA, 1.5 mM $MgCl_2$) supplemented with cOmplete protease inhibitor cocktail (39802300, Roche) and orthovanadate (Sigma-Aldrich). After assessing protein concentrations using the Bradford method (5000006, Bio-Rad Laboratories), equal amounts of total protein were loaded onto SDS-PAGE, and proteins were transferred to PVDF membranes (10600023, Cytiva). The membranes were then incubated with 5% bovine serum albumin (BSA; 04-100-812-D, Euromedex) in TBS-Tween and probed with the following primary antibodies overnight at 4 °C: ERK2 (sc-1647, Santa Cruz, 1:1000), LSD1 (2139S, Cell Signaling Technology, 1:1000), or CDK6 (3136,

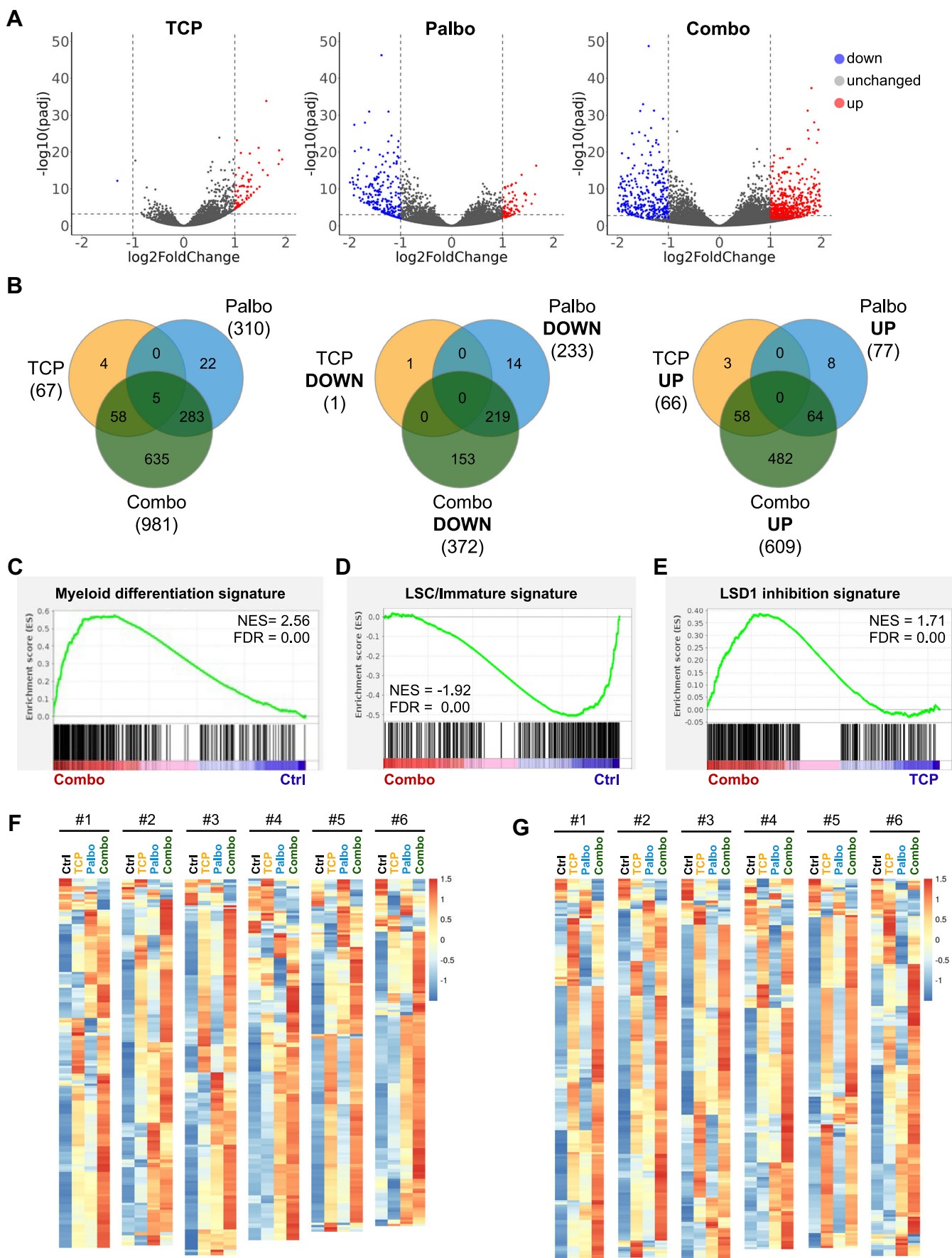

**Figure 6. Transcriptomic profiling of primary patient samples supports the loss of the immature compartment and increased LSD1 inhibition by the combination.**

RNA-seq analysis of 6 primary AML human samples treated either with the combination (0.5 μM palbociclib and 5 μM TCP) or the single molecules (TCP or Palbo) for 96 h. (A) Volcano plots illustrating the significantly deregulated genes for each treatment compared to the control cells (log2 fold change > 1, $p_{adj}$ < 0.01) of primary patient samples ($n = 6$). Statistical significance was assessed using Wald test, and the resulting $p$ values were adjusted for multiple testing using the Benjamini–Hochberg procedure to control the false discovery rate. (B) Venn diagrams indicating the number of significantly deregulated genes (left panel), and the number of down- and up-regulated genes compared to the control cells (middle and right panels). The thresholds were log2 fold change > 1, and $p_{adj}$ < 0.01. (C, D) GSEA of myeloid differentiation (C) and LSC/immature signatures (D) comparing the combination treatment to the control. (E) GSEA of the LSD1 signature, comparing the combination treatment to TCP. (F, G) Heatmaps of the differentiation signature (F) and the LSD1 signature (G) for each individual patient sample. Normalized TPM values are represented as Z-scores.

Cell Signaling Technology, 1:2000). Following incubation with HRP-conjugated anti-mouse (NA931, Cytiva) or anti-rabbit (NA934, Cytiva) secondary antibodies, blots were visualized using ECL Prime (RPN2236, Cytiva) and ImageQuant 800 (Cytiva). Band intensities were quantified using ImageJ. Each band quantity was calculated by measuring band intensity minus background. Total proteins were normalized based on ERK2 levels and calculated relative to the control sample.

## May-Grünwald–Giemsa staining

About $5 \times 10^4$ AML cells were centrifuged for 5 min at 1000 rpm onto SuperFrost microscopic slides (12302108, Thermo Fisher Scientific) and then stained with May-Grünwald–Giemsa (MGG) solution as previously described (Devillier et al, 2015). Images were captured using an Apotome microscope (Zeiss) and analyzed with ImageJ software.

## Colony-forming cell (CFC) assays

About $1–3 \times 10^3$ treated AML cell lines or $1 \times 10^5$ treated AML patient cells were thoroughly mixed with 1 mL of methylcellulose medium (MethoCult H4230 or H4535, StemCell Technologies, respectively), dispensed into 35 mm cell culture dishes, and incubated for 10 days at 37 °C. Colonies were then quantified using MTT staining (M5655, Sigma-Aldrich). Images were captured using an optical binocular magnifier and analyzed with the colony-counter plugin of ImageJ software. To assess serial replating capacity, colonies were resuspended in IMDM medium, and again, $1–3 \times 10^3$ progenitor cells from AML cell lines or $1 \times 10^5$ progenitors from patient samples were replated in MethoCult, as described above. Rounds of serial replating were performed until progenitors from one treatment condition were exhausted.

## Flow cytometry

For progenitor assessment, fresh murine bone marrow cells were extracted, and depletion of erythrocytes was performed using a red blood cell lysis buffer as per the manufacturer's instructions (07850, Stem Cell Technologies). Then, lineage-positive cells were stained with a lineage depletion kit (130-090-858, Miltenyi Biotec, 1:4) and a PE-CF594-streptavidin-secondary antibody (562284, BD Biosciences, 1:400). For progenitor analysis, the following antibodies were used: CD34-VioGreen (130-124-460, Miltenyi Biotec, 1:100), Sca-1-PerCP-Cy5.5 (122524, Biolegend, 1:400), c-Kit-APC-eFluor780 (47-1171-82, Invitrogen, 1:400), CD16/32-PE-Cy7 (25-0161-82, Invitrogen, 1:1000) and CD45-AF700 (103128, Biolegend, 1:400). For dead cell discrimination, Sytox Blue (S34857, Thermo Fisher Scientific, 1:20,000) was used.

For engraftment monitoring, following red blood cell lysis, cells were stained with LIVE/DEAD stain kit (L10120, Far Red, Thermo Fisher Scientific, 1:200), and labeled with human CD45-Pacific Blue (304029, Biolegend, 1:50), mouse CD45-APC (47-0451-82, Life Technologies, 1:200) and human CD33-PE (555450, BD Pharmingen, 1:30).

For analysis of differentiation markers, cells were harvested, stained with LIVE/DEAD stain kit (L10120, Far Red, Thermo Fisher Scientific, 1:200), washed in PBS with 3% FBS, and resuspended in PBS/FBS. Cells were incubated with CD11b-PE (IM2581U, Beckman Coulter, 1:300), CD14-BV510 (367124, BioLegend, 1:200), CD86-V450 (560357, BD Biosciences, 1:50), or CD163-FITC (563697, BD Pharmingen, 1:50) antibody.

For DNA content analysis, cells were fixed and permeabilised (554714, BD Biosciences), and then stained with DAPI (D9542, Sigma-Aldrich).

For cell proliferation assays, the CellTrace CFSE (C34554, Thermo Fisher Scientific) was used following the manufacturer's protocol. In brief, $2 \times 10^6$ cells were incubated with 5 μM CFSE at 37 °C for 10 min.

For cell death assays, the AnnexinV Apoptosis Detection kit (559763, BD Biosciences) was used according to the manufacturer's instructions. The ferroptosis assay was performed using the lipid peroxidation sensor BODIPY 581/591 C11 (D3861, Thermo Fisher Scientific).

All acquisitions were performed on BD Fortessa or BD LSR II cytometers, and analyzed using FlowJo v10 software.

## Drug synergy assay

Synergy assays were performed on MV4-11 cells using the following compounds: palbociclib at 0.125, 0.25, 0.5, and 1 μM; ribociclib at 0.25, 0.5, 1, and 2 μM; abemaciclib at 6.25, 12.5, 25, and 50 nM; TCP at 1.25, 2.5, 5, and 10 μM; ORY-1001 at 0.125, 0.25, 0.5, and 1 nM; and GSK2879552 at 1.25, 2.5, 5, and 10 μM. CD11b expression levels were analyzed by flow cytometry. Synergy scores were assessed with the SynergyFinder web application (Ianevski et al, 2020), and synergy distribution was represented with a ZIP-based model. A ZIP synergy score $<-10$ indicates an antagonistic interaction, scores from $-10$ to 10 suggest an additive interaction, and scores >10 indicate a synergistic interaction.

## Animal experiments

All animal studies were performed according to protocols approved by the French National Committee on Animal Care (APA-FIS#33898-2021110816214653 for both xenotransplantation and toxicity assays, and APAFIS#21269-2019062814106696 for PDX

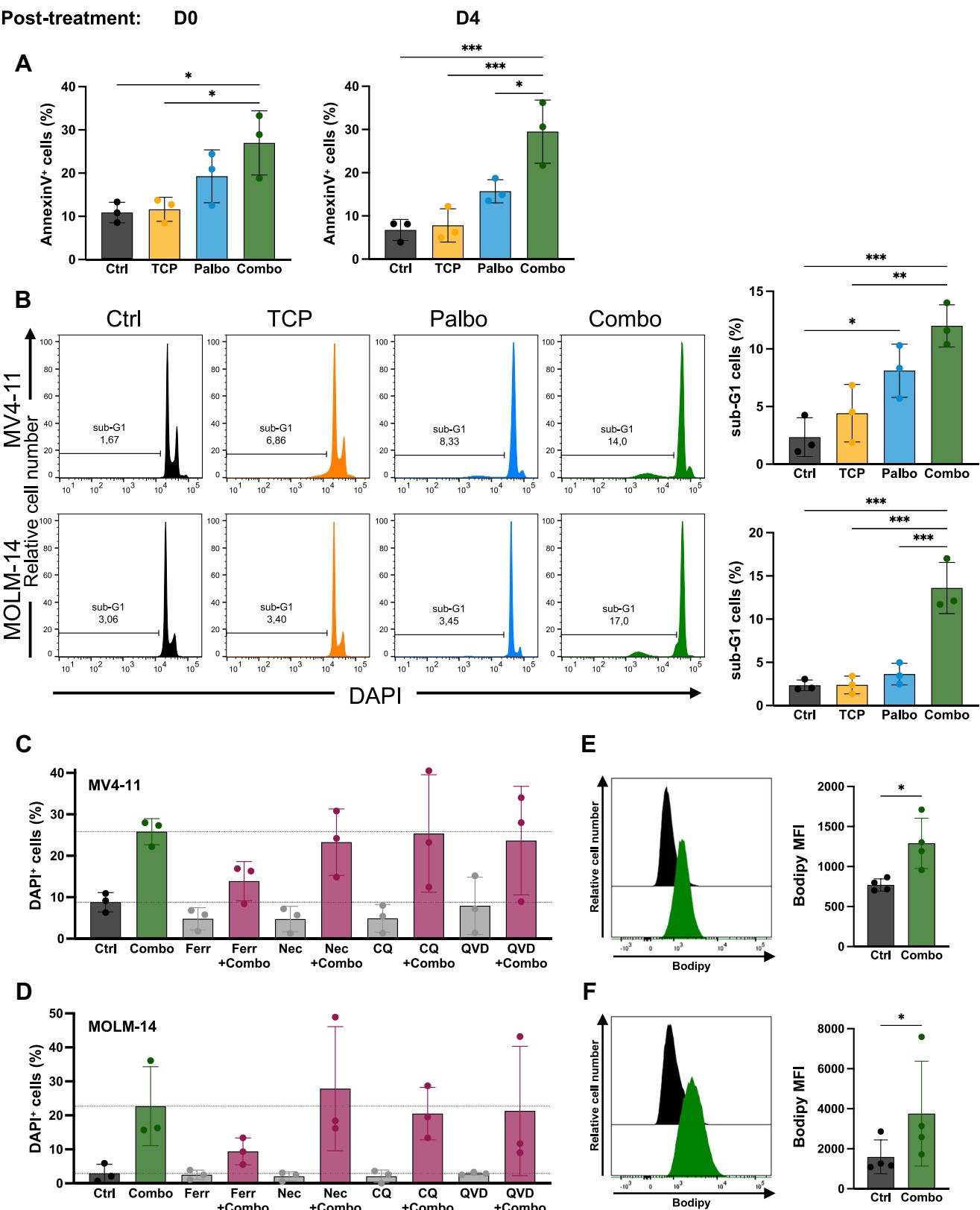

**Figure 7.  The combined treatment induces ferroptosis-dependent cell death.**

(A) Quantification of cell death using Annexin V staining on MV4-11 cells treated for 96 h as previously described. At the end of the treatments, cells were harvested for cell death analysis using Annexin V-DAPI staining and flow cytometry. Additionally, a separate aliquot of cells was maintained in culture for 4 days, followed by the cell death analysis. Data from $n = 3$ independent experiments are represented as mean ± SD. Statistical analyses were performed using a one-way ANOVA followed by Tukey's test. *$p < 0.05$, ***$p < 0.001$. Left panel, Ctrl vs Combo $p = 0.0284$, TCP vs Combo $p = 0.0348$. Right panel, Ctrl vs Combo $p = 0.0007$, TCP vs Combo $p = 0.0010$, Palbo vs Combo $p = 0.0101$. (B) Analysis of DNA content following the indicated treatments. MV4-11 and MOLM-14 cells were permeabilized and stained with DAPI. Sub-G1 cells were quantified by flow cytometry ($n = 3$). One representative experiment out of three is shown (left panel), with quantifications (right panel). Data represent the mean ± SD. Statistical analyses were performed using a one-way ANOVA followed by Tukey's test. *$p < 0.05$, **$p < 0.01$, ***$p < 0.001$. Top panel, Ctrl vs Palbo $p = 0.0110$, Ctrl vs Combo $p = 0.0008$, TCP vs Combo $p = 0.0028$. Bottom panel, Ctrl vs Combo $p = 0.0003$, TCP vs Combo $p = 0.0003$, Palbo vs Combo $p = 0.0006$. (C, D) MV4-11 (C) and MOLM-14 cells (D) were treated with the vehicle control (Ctrl) or the combination (Combo), in the presence of the indicated inhibitors. DAPI staining was performed and quantified by flow cytometry (mean ± SD of $n = 3$ independent experiments). Ferr ferrostatin-1, Nec necrostatin-1, CQ chloroquine, Q-VD-OPH caspase inhibitor. (E, F) Quantification of lipid oxidation using C11-BODIPY (581/591) control MV4-11 (E) and MOLM-14 (F) cells, treated with vehicle control (black histograms) or with the combination (green histograms). Data represent the mean ± SD of $n = 4$ independent experiments. *$p < 0.05$, paired one-tailed Student's $t$-test. Exact $p$ values were 0.0134 in (E) and 0.0471 in (F).

establishment). NOD.Cg-Prkdc scid/J (NSG) mice were obtained from Charles River France or bred in-house and maintained on a 12-h light and 12-h dark cycle, under specific pathogen-free conditions with sterilized food and water provided ad libitum. The animals were housed collectively in disposable standard cages, placed in ventilated racks.For mouse survival experiments, MV4-11 were treated for 4 days in vitro with vehicle control (DMSO), ATRA (5 µM), TCP (5 µM), Palbociclib (0.5 µM) alone or with the combination Palbociclib/TCP. Approximately $10^5$ cells in PBS were injected into the tail vein of 6–8-week-old male NOD-SCID-common-γ-chain-knockout (NSG) mice (Charles River). Mice were monitored daily for body weights and behavior. Humane endpoints included weight loss >20%, ruffled coat, hunched back and reduced motility.

For cell line xenografts, stable luciferase-expressing MOLM-14-luciferase cells were generated by a lentiviral infection of a pRRL.GFP.luc2 vector. Approximately $2 \times 10^5$ cells resuspended in 100 µL PBS were injected into the tail vein of 6–8-week-old male NOD-SCID-common-γ-chain-knockout (NSG) mice (Charles River). One day post-transplantation, bioluminescence was measured to evenly assign mice into four groups: vehicle control (PBS), ORY-1001 (0.0125 mg/kg), palbociclib (25 mg/kg) and combination ORY-1001+palbociclib groups. Treatments were administered intraperitoneally on a 4-day ON/3-days OFF schedule for 4 weeks. Mice were monitored daily for body weights and behavior, and bioluminescence imaging was performed once a week. Humane endpoints included weight loss >20%, ruffled coat, hunched back and reduced motility.

For the establishment of PDX models, primary AML patient samples were thawed in 10 mL IMDM medium supplemented with 20% heat-inactivated FBS. Cells were washed twice with cold PBS, and $10^7$ cells were resuspended in a mix of 80 µL autoMACS Running Buffer—MACS Separation Buffer (130-091-221, Miltenyi Biotec) and 20 µL CD3 MicroBeads (130-050-101, Miltenyi Biotec). The suspension was gently mixed, incubated for 15 min at 4 °C, and washed with 2 mL autoMACS Running Buffer. Cells were centrifuged at $300 \times g$ for 10 min, resuspended at $10^9$ cells/mL in autoMACS Running Buffer, and magnetic separation was performed using autoMACS Pro Separator under sterile conditions, according to the manufacturer's instructions.

About $10^6$ AML patient cells, depleted from CD3 and resuspended in 100 µL PBS, were injected into the tail vein of sub-lethally irradiated (1.5 Gy) 6–10-week-old NSG mice (Charles

River). Engraftments were monitored in peripheral blood by flow cytometry. Mice were sacrificed when a humane endpoint was reached. For secondary and tertiary transplantations, the same procedure was applied. Cells from bone marrow and spleen were collected to establish a frozen collection of AML PDX samples.

For PDX treatment, $10^5$–$10^6$ cells were injected intravenously into the tail vein of 6–10-week-old male NSG mice, and engraftments were monitored by measuring circulating hCD45+/hCD33+ blasts in peripheral blood. Once hCD45+/hCD33+ blasts were detected in the bloodstream, mice were allocated to treatment groups -vehicle control (PBS), ORY-1001 (0.0125 mg/kg), palbociclib (25 mg/kg), and combination ORY-1001+palbociclib- and the treatments started. Treatments were administered intraperitoneally on a 4-day ON/3-days OFF schedule for 3–5 weeks. Mice were monitored daily for body weights and behavior. Humane endpoints included weight loss >20%, ruffled coat, hunched back and reduced motility. Mice were sacrificed when they reached a humane endpoint, and bone marrow and spleen were collected for analysis.

For assessment of the treatment on normal hematopoiesis, ORY-1001+palbociclib (Combo) treatment was administrated intraperitoneally to 8-week-old adult male C57BL/6J immunocompetent mice (Charles River) on a 4-days ON/3-days OFF schedule. Blood analyses were performed after 28 days of treatment using a hematology analyzer (OV-360, Balio Diagnostics).

All experiments were conducted using male mice, with the exception of the data presented in Appendix Fig. S1D, which utilized female mice. The age of mice at the start of each experimental series was: 6–8 weeks for Fig. 3A–C, 9–10 weeks for Fig. 3D PDX1, 8–9 weeks for Fig. 3D PDX2, 11 weeks for Fig. 3D PDX3, 10–11 weeks for Fig. 3D PDX4, 10 weeks for Fig. 3D PDX5, 8–9 weeks for Fig. 3E, 10–11 weeks for Fig. 3F, 6–8 weeks for Fig. EV4, 7–8 weeks for Appendix Fig. S1D, and 10–11 weeks for Appendix S1E.

## RNA-seq analyses

Total RNA was isolated using the RNAeasy kit (74104, Qiagen) following the manufacturer's instructions. RNA libraries were prepared using the Illumina Stranded mRNA Prep kit (20040532), and sequencing was performed by the TGML (Marseille, France), and the iGenSeq (Paris, France) facilities on an Illumina NextSeq 500. The primary AML samples used for RNA-seq analysis were amplified in NSG mice for at least three passages to increase the

number of leukemic cells. These PDX samples were re-sequenced to confirm the persistence of the mutations identified in the original samples.

Sequencing quality control was determined using FastQC (0.11.9) tool (http://www.bioinformatics.babraham.ac.uk/projects/fastqc/). Replicate concurrency was verified using correlation heatmap (Pearson correlation) from Deeptools suite (v3.2.1), and sample replicates were merged into a single bam file.

Reads were mapped to the hg19 UCSC genome using subread-align (v1.6.4) (Liao et al, 2013) with default parameters. Gene expression was determined by counting mapped tags at gene levels using featureCounts (Liao et al, 2014). The normalization and differential expression analysis were performed with the R/Bioconductor (DESeq2, estimateSizeFactors function) v.1.26.0 (Love et al, 2014). Counts for differentiating genes were then converted to transcripts per million (TPM) for each sample, and z-scores were calculated for each gene. Hierarchical clustering was performed using Euclidean distance. The Gene Ontology (GO) biological processes associated with candidate genes were determined using the clusterprofiler v4.6.0 tool with a p value <0.05 and by taking into account a background (clusterProfiler 4.0: A universal enrichment tool for interpreting omics data). For GSEA analysis, we used GSEA software on the Java platform (v4.0.3) (Subramanian et al, 2005). The myeloid differentiation signature was a mix of five signatures and contained 382 genes (incl. GO:0045655, GO:0045657) (Brown et al, 2006; Song et al, 2019; Coillard and Segura, 2019); the immature myeloid AML signature was a mix of three published signatures, and contained 395 genes (Pabst et al, 2016; Eppert et al, 2011; Somervaille et al, 2009; Gal et al, 2006); the response to LSD1 inhibition signature in AML contained 375 genes (Maes et al, 2018; Sugino et al, 2017; Smitheman et al, 2019); the SPI1 downstream gene UP and DOWN signatures contained 1349 and 1779 genes, respectively (McKenzie et al, 2019); the apoptosis signature contained 161 genes (GSEA HALLMARK_APOPTOSIS, M5902) (Liberzon et al, 2015). The ferroptosis signature contained 64 genes (GSEA WP_FER-ROPTOSIS, M39768).

## ATAC-seq analyses

MV4-11 cells were harvested for DNA tagmentation and library preparation using the ATAC-Seq Kit (53150, Active Motif), following the manufacturer's instructions. sequenced at the TGML sequencing facility on an Illumina NextSeq 500 with a flow cell providing 300M reads and 150 cycles.

Sequencing quality control was determined using FastQC (0.11.9) tool (http://www.bioinformatics.babraham.ac.uk/projects/fastqc/). Replicate concurrency was verified using correlation heatmap (Pearson correlation) from Deeptools suite (v3.2.1), and sample replicates were merged into a single bam file.

Reads were mapped to the *Homo sapiens* (hg19) genome as previously described (Orlando et al, 2014) using default parameters of Bowtie2 (v2.3.4.3) (Langmead and Salzberg, 2012). Duplicate tags were marked and removed using Picard MarkDuplicates (v2.20.2), and mapped tags were processed for further analysis. High confidence binding sites were determined using MACS2 peak caller, sharp mode (-p 0.01) (Zhang et al, 2008). For quantitative analysis, hg19 mapped tags were counted within peak coordinates (defined previously by MACS2) using featureCounts (Zhang et al,

### The paper explained

#### Problem

Acute myeloid leukemia (AML) is an aggressive malignancy characterized by the accumulation of myeloid blasts in the bone marrow, which leads to hematopoiesis deficiency. AML is very heterogeneous with diverse genetic landscapes and complex intra-patient clonality that challenge treatment response, influence disease progression and ultimately affect prognosis. Standard chemotherapy is effective in fit patients, but its effects are often transient and largely insufficient for those with high-risk genetic mutations or for the elderly. Leukemic cells are more plastic than normal cells, and ultimately, some cancer cells develop resistance. While several therapeutic targets have been recently identified, they have proven insufficient as monotherapies or in combination with chemotherapy.

#### Results

Our work focused on CDK6, a vulnerability present in several subtypes of AML that can be targeted using molecules approved for breast cancer treatment. By combining CDK6 inhibitors with existing molecules of interest in leukemia, we discovered a two-molecule regimen that restores leukemic blast differentiation and demonstrates efficacy in leukemia samples in the laboratory, including those from patients who have relapsed. In addition, the combination therapy targets the small population of leukemic progenitors, and induces cell death. At the molecular level, the combination remodels chromatin accessibility, thereby impairing the imprinting of immature leukemic cells. Interestingly, unlike the single molecules, the combination induces long-lasting effects on the leukemic cells. Finally, in preclinical mouse models, the combination therapy reduces leukemic burden in various genetically distinct primary AML samples.

#### Impact

Here, we demonstrate that repositioning of CDK4/6 inhibitors in AML can enhance the efficacy of drugs currently in clinical trials (LSD1 inhibitors). Furthermore, this combination may offer a viable therapeutic option in cases of resistance to conventional therapy. Finally, since the combination shows synergy, these molecules could be used at lower concentrations, potentially leading to fewer side effects. This study provides valuable insights into the potential repositioning of existing FDA-approved drugs for use in AML.

2008). For IGV visualization, mapped tags were converted into BigWig using the deepTools suite (v3.2.1) (bamCoverage, multi-BigwigSummary) (Ramírez et al, 2016). In order to merge the peaks, we used HOMER mergePeaks (v 4.11). For heatmaps and profiles, we used deepTools (v3.5.1) to generate read abundance from all datasets around the peak center (–2.0 kb/2.0 kb), using "computeMatrix". These matrices were then used to generate heatmaps and profiles, using deepTools commands "plotHeatmap" or "plotProfile", respectively. De novo binding motif analysis was performed on ATAC-peaks using HOMER (v.4.11).

## Statistical analyses

Measurements were presented as the mean and standard deviation (SD) of each condition, unless specified. Student t-test were applied when only two groups were compared. Unless stated, t-test were paired to take into account the measures made within the same experiment. For the analyses of more than two groups, the comparisons were done using one-way analysis of variance (ANOVA) followed by Tukey's multiple comparison test.

Significant thresholds are indicated by asterisks as follows: 0.05 (*), 0.01 (**), or 0.001 (***). For the mouse survival analysis, the Kaplan–Meier method and the log-rank Mantel–Cox test were applied. *P* values of less than 0.05 were considered statistically significant. All statistical analyses were performed using GraphPad Prism v5 software.

Transplanted mice were randomly assigned to treatment groups, ensuring comparable leukemia cell counts prior to drug administration across groups. Preliminary experiments were conducted to estimate the sample size required per treatment group to achieve 80% power in detecting a 50% difference in leukemic growth reduction. Blinded assessment was conducted to ensure unbiased analysis of the mice. No animals were excluded from the analyses.

# Data availability

RNA-Seq data from MV4-11 cell model: Gene Expression Omnibus GSE297639. RNA-Seq data from primary AML: Gene Expression Omnibus GSE297638. ATAC-Seq data: Gene Expression Omnibus GSE297641. Source Data for EMM-2024-20757 has been assigned the BioStudies accession number S-BSST2077.

The source data of this paper are collected in the following database record: biostudies:S-SCDT-10_1038-S44321-025-00296-2.

# Peer review information

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

## Acknowledgements

The authors thank all members of the De Sepulveda laboratory. We are grateful to M Richaud and F Mallet for assistance with the use of the flow cytometry facility. We thank C Montersino, E Josselin, M Macario, and C Bonnet from the TrGet platform for their help with animal experiments. We thank E Delabesse, V De Mas (HIMIP CRB), and the CRCM-IPC biobank, who kindly provided patient samples, and sincerely thank all those patients, and their families, who have accepted to provide samples and information. The authors thank the hematology department of Institut Paoli-Calmettes for their help with staining and immunophenotyping patient samples. The authors thank N Platet for her help in immunophenotyping. The authors thank J Adelaide and M Chaffanet for providing sequencing data on patient samples. The authors also wish to acknowledge their gratitude to the animal support facility of the CRCM, the microscopy platform, and the CIBI and DISC platforms for computational analyses and support. We thank the Canceropôle PACA, IBISA and the Plan Cancer Equipement (#17CQ047-00) for continued support in the development of the CRCM TrGET preclinical assay platform. High-throughput sequencing was performed at the TGML Platform, supported by grants from Inserm, GIS IBiSA, Aix-Marseille Université, and ANR-10-INBS-0009-10, and from equipment and services from the iGenSeq core facility at ICM, Paris. This work was supported by Canceropôle Provence-Alpes-Côte d'Azur, the French National Cancer Institute (INCa), the Provence-Alpes-Côte d'Azur Region (PDS), Gefluc (S. Lo), Fondation ARC (PDS), the patient associations Fédération Leucémie Espoir (PDS), Association Cassandra (LB and PDS), the Société Française d'Hématologie, SFH (LB), Fondation de France (EV), and the EU H2020 MSCA program (EV).

## Author contributions

**Lise Brault**: Formal analysis; Validation; Investigation; Visualization; Methodology; Writing—original draft; Writing—review and editing. **Edwige Voisset**: Funding acquisition; Validation; Visualization; Writing—original draft; Writing—review and editing. **Mathieu Desaunay**: Investigation. **Antonia Boudet**: Investigation. **Paraskevi Kousteridou**: Investigation; Visualization; Writing—original draft. **Sébastien Letard**: Investigation. **Nadine Carbuccia**: Investigation. **Armelle Goubard**: Investigation. **Rémy Castellano**: Resources; Methodology. **Yves Collette**: Resources; Methodology. **Julien Vernerey**: Data curation; Software; Writing—original draft. **Isabelle Vigon**: Resources. **Jean-Max Pasquet**: Resources. **Patrice Dubreuil**: Resources; Funding acquisition. **Sophie Lopez**: Conceptualization; Resources; Formal analysis; Validation; Investigation; Methodology; Writing—original draft; Writing—review and editing. **Paulo De Sepulveda**: Conceptualization; Resources; Formal analysis; Supervision; Funding acquisition; Methodology; Writing—original draft; Project administration; Writing—review and editing.

Source data underlying figure panels in this paper may have individual authorship assigned. Where available, figure panel/source data authorship is listed in the following database record: biostudies:S-SCDT-10_1038-S44321-025-00296-2.

## Disclosure and competing interests statement

The authors declare no competing interests.

# Expanded View Figures

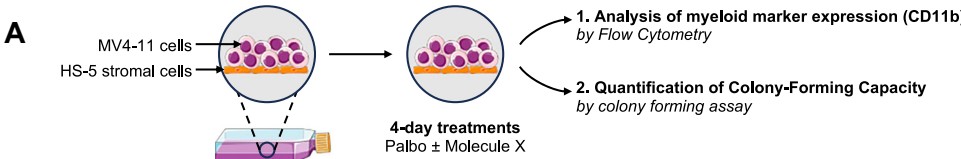

**A**

MV4-11 cells
HS-5 stromal cells

4-day treatments
Palbo ± Molecule X

1. **Analysis of myeloid marker expression (CD11b)**
   *by Flow Cytometry*

2. **Quantification of Colony-Forming Capacity**
   *by colony forming assay*

**B**

CD11b (%)

| | Molecule alone | Molecule in combination with palbociclib |
|---|---|---|
| *Vehicle control* | 4,2 | - |
| *Palbociclib* | 21,6 | - |
| 8-CPT | 4,2 | 13,8 |
| AC220 | 3,4 | 10,7 |
| AGI5798 | 0,7 | 5,5 |
| AGI6780 | 0,6 | 5,3 |
| Arsenic trioxide | 2,3 | 20,7 |
| ATRA | 11,1 | 51,3 |
| Azacitidine | 3,1 | 5 |
| Calcitriol | 32,5 | 74,3 |
| Decitabine | 10,4 | 7 |
| Entinostat | 1,8 | 10,6 |
| OTX015 | 2,8 | 12,1 |
| Panobinostat | 1,6 | 9,8 |
| Prima-1Met | 0,2 | 10 |
| Ruxolitinib | 0,2 | 2 |
| SGI110 | 7,4 | 5,8 |
| TCP | 9,6 | 44,7 |
| Valproic acid | 35,8 | 41,3 |
| Venetoclax | 3,8 | 12,3 |
| Vorinostat | 0,9 | 6,7 |

**C**

| Treatment | No. of CFU-L | % CFU-L |
|---|---|---|
| Control | 277 | 100.0 |
| Palbo | 78 | 28.2 |
| TCP | 282 | 101.8 |
| Palbo+TCP | 27 | 9.7 |
| ATRA | 218 | 78.7 |
| Palbo+ATRA | 270 | 97.5 |

**Figure EV1.  Screening of 19 molecules in combination with palbociclib.**

(A) Experimental design for the drug screening: MV4-11 cells cocultured with HS-5 stromal cells were pretreated for 24 h with either vehicle control or palbociclib, followed by the exposure to either the second molecule alone, or the combination of palbociclib and the second molecule for an additional 72 h. Cells were then harvested to assess the expression of the myeloid differentiation marker CD11b by flow cytometry and to quantify the minor compartment of colony-forming cells. (B) List of molecules used in the screen (first column), and results of the analyses of CD11b expression following monotherapies (second column) or combined treatments with palbociclib (third column). Heatmap was applied to highlight the increase in CD11b expression. (C) Results of the colony-forming cell assay for monotherapies involving TCP and ATRA or in combination with palbociclib. Colonies (CFU-L) are expressed as numbers (second column) and as the percentage of colonies relative the control cells (last column). Heatmap was used in the table to highlight the drop in colony number.

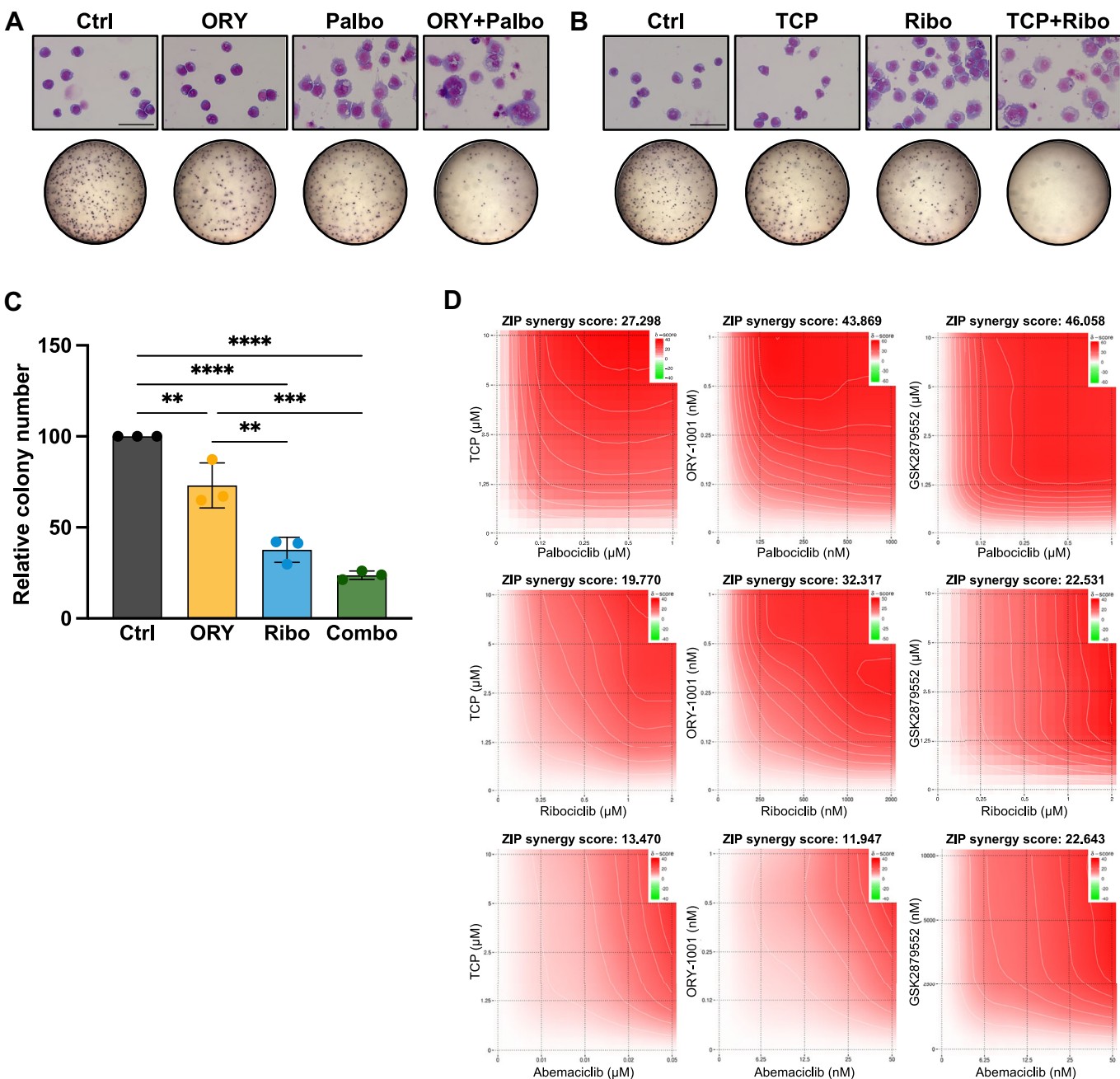

**Figure EV2. Dual targeting of CDK6 and LSD1 is synergistic.**

(A, B) Alternative LSD1 or CDK6 inhibitors. MV4-11 cells were treated with the LSD1 inhibitor ORY-1001 (ORY) with palbociclib (A) or with TCP and the CDK6 inhibitor ribociclib (Ribo) (B). After the 4-day treatment, cells were stained with May-Grünwald–Giemsa (top panel), and their capacity to form colonies was evaluated (bottom panel). (C) MV4-11 cells cocultured with HS-5 stromal cells were incubated with CDK6 inhibitor ribociclib at 1 μM, with ORY-1001 at 0.5 nM, or with the combination of ribociclib and ORY-1001 (Combo) for 96 h. After the treatment, leukemic cells were seeded in equal numbers in methylcellulose for 10 days. Data represent the mean ± SD of n = 3 independent experiments. Statistical analyses were performed using a one-way ANOVA followed by Tukey's test. **p < 0.01, ***p < 0.001, ****p < 0.0001. Adjusted pvalues: Ctrl vs ORY p = 0.0075, Ctrl vs Ribo p < 0.0001, Ctrl vs Combo p < 0.0001, ORY vs Ribo p = 0.0018, ORY vs Combo p = 0.0003. (D) Evaluation of the synergy of CDK6 and LSD1 inhibitor combinations. MV4-11 cells were treated with increased concentrations of CDK6 inhibitor (palbociclib 0.125 to 1 μM; ribociclib 0.25 to 2 μM; abemaciclib 6.25 to 50 nM) in combination with LSD1 inhibitor (TCP 1.25 to 10 μM; ORY-1001 0.125 to 1 nM; GSK2879552 1.25 to 10 μM) for 96 h. The percentage of cells expressing CD11b was quantified by flow cytometry. Results were analyzed using the SynergyFinder tool, and represented by 2D contour plots. The red color indicates synergy.

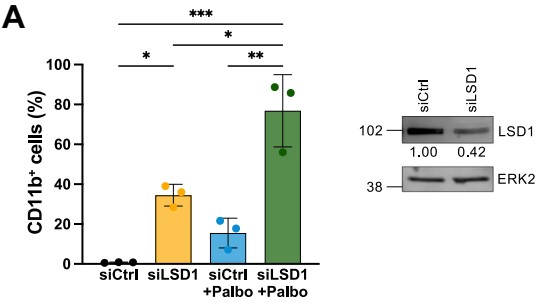

**A**

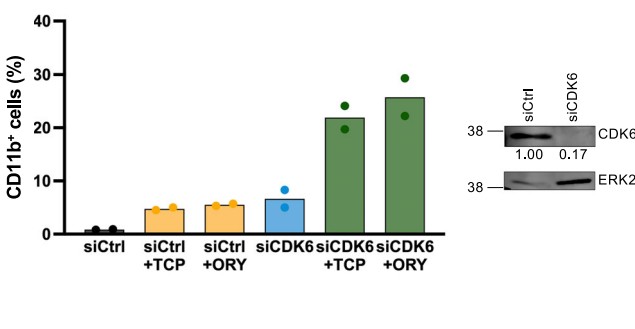

**B**

**C**

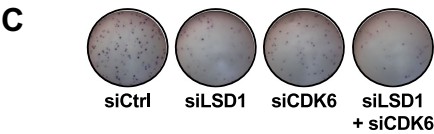

**Figure EV3.   Targeting of LSD1 and CDK6 by RNA interference recapitulated results obtained with small-molecule inhibitors.**

(A) MV4-11 cells transfected with control (Ctrl) or LSD1 siRNAs were incubated with palbociclib (0.5 μM), as indicated, for 4 days. The percentage of CD11b-expressing cells was assessed by flow cytometry (left panel). Data represent the mean ± SD of $n = 3$ independent experiments. LSD1 protein expression was quantified by Western blot analysis (right panel); one of three experiments is shown. Statistical analyses were performed using a one-way ANOVA followed by Tukey's test. *$p < 0.05$, **$p < 0.01$, ***$p < 0.001$. Adjusted pvalues: siCtrl vs siLSD1 $p = 0.0363$, siCtrl vs siLSD1+Palbo $p = 0.0006$, siLSD1 vs siLSD1+Palbo $p = 0.0131$, siCtrl+Palbo vs siLSD1+Palbo $p = 0.0020$. (B) MV4-11 cells transfected with control (Ctrl) or CDK6 siRNAs were incubated with 5 μM TCP or 0.5 nM ORY-1001, as indicated, for 4 days. The percentage of CD11b-expressing cells was assessed by flow cytometry (left panel). Data represent the mean ± SD of $n = 2$ independent experiments. CDK6 protein expression was quantified by Western blot. One representative experiment out of two is shown (right panel). (C) MV4-11 cells transfected with control, CDK6 and/or LSD1 siRNAs were seeded on methylcellulose. CFCs were analyzed at day 10 post plating.

## A. C57BL/6

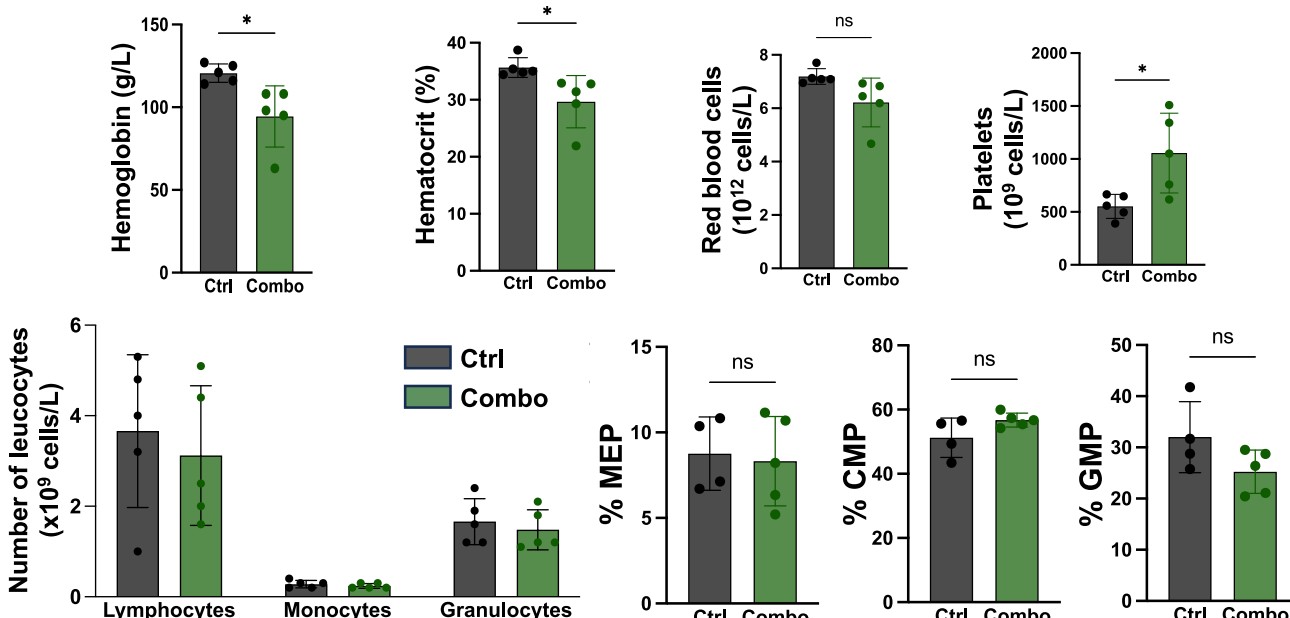

## B. PDX

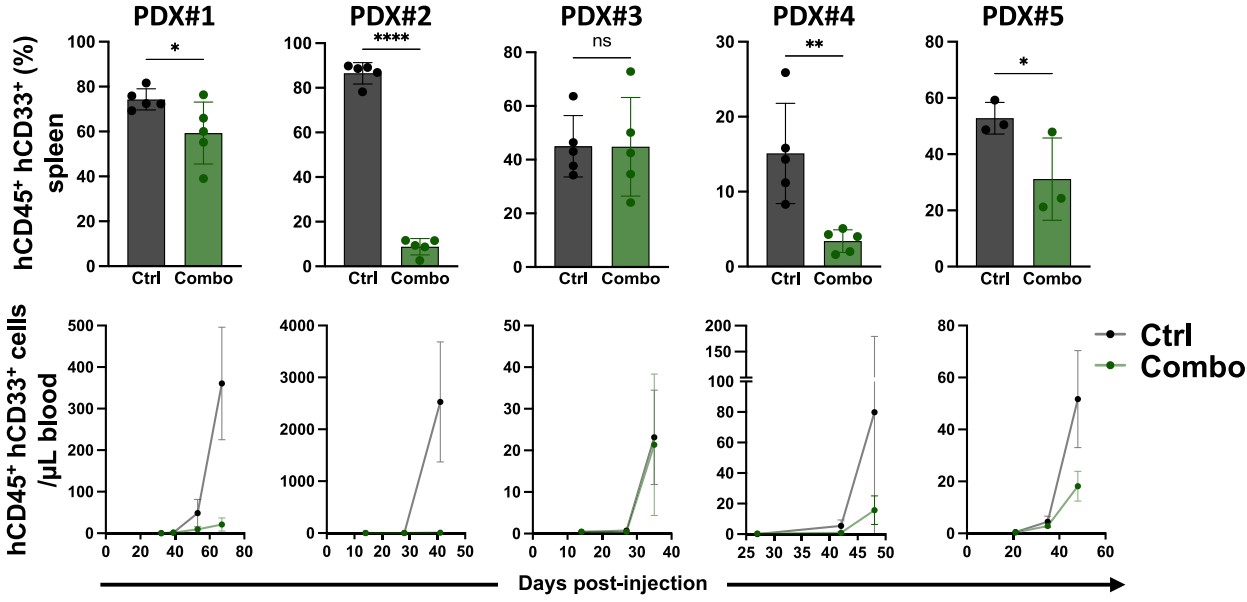

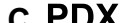

## C. PDX

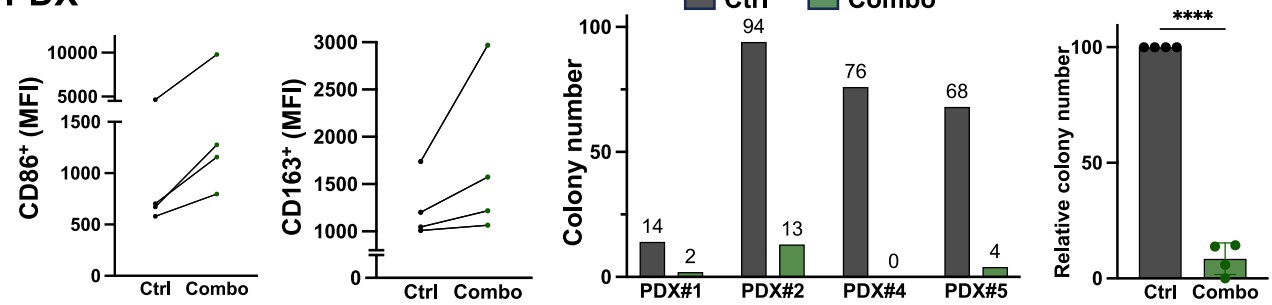

◀ **Figure EV4. Analyses of in vivo effects of the combination on C57BL/6 and AML PDX mice.**

(A) Evaluation of the combined treatment on mouse hematopoiesis. C57BL/6 mice were treated with intraperitoneal injections of the vehicle control or a combination of palbociclib (25 mg/kg) and ORY-1001 (0.0125 mg/kg), 4 days a week for 4 weeks. At the end of the treatments, blood counts and the bone marrow myeloid progenitors were analyzed. Data represent the mean ± SD of $n = 5$ mice. Statistical analysis was performed using an unpaired two-tailed Student's $t$-test. ns is not significant. *$p < 0.05$. CMP common myeloid progenitors, GMP granulocyte-monocyte progenitors, MEP megakaryocyte-erythroid progenitors. Top panels, $p = 0.0163$, $p = 0.0254$, $p = 0.0519$, and $p = 0.0211$. Bottom panels, $p = 0.6117$, $p = 0.4861$, $p = 0.8805$, $p = 0.7955$, $p = 0.0995$, and $p = 0.1129$. (B) Quantification of hCD45 + hCD33+ leukemic blast cells in the spleen and blood of the PDX NSG mouse models, as shown in Fig. 3D. PDX#1 $n = 5$, PDX#2 $n = 5$, PDX#3 $n = 5$, PDX#4 $n = 5$, PDX#5 $n = 3$. Statistical analysis was performed using an unpaired one-tailed Student's $t$-test. *$p < 0.05$, **$p < 0.01$, and ****$p < 0.0001$. (C) Analysis of the leukemic cells remaining following in vivo treatment of PDX mouse models. (Left) The expression of CD86 and CD163 late myeloid markers were analyzed on the remaining leukemic cells (detected as hCD45+ cells), for each untreated and matched PDX treated pair. (Middle and Right) Evaluation of the number of CFC progenitors remaining in the bone marrow of four independent PDX samples. Cells were seeded in methylcellulose in equal numbers for 10 days. The right histogram combined results from the four PDX, expressed as a percentage of colonies obtained relative to the control cells. Data represent the mean ± SD of $n = 4$ samples per condition. Statistical analysis was performed using a paired one-tailed Student's $t$-test, ****$p < 0.0001$.

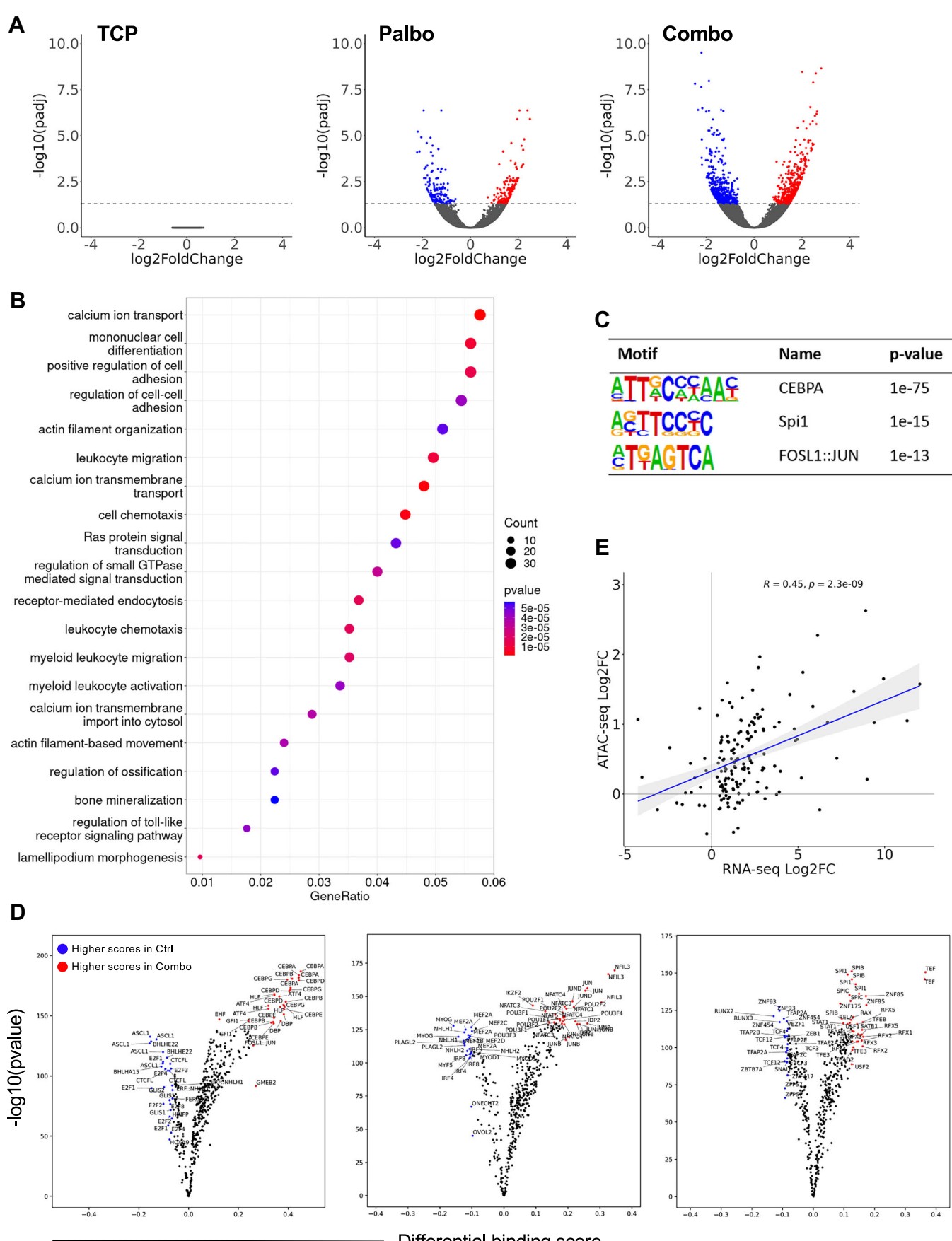

◀ **Figure EV5. Analysis of chromatin accessibility of MV4-11 cells.**

(A) Volcano plot representing modifications of chromatin accessibility assessed by ATAC-seq ($p_{adj} < 0.05$), of MV4-11 cells treated with the combination or the single molecules, compared with the control cells, as indicated ($n = 2$). (B) Gene Ontology biological processes modified by the combination treatment, corresponding to significantly deregulated chromatin regions. Statistical analysis was done using a hypergeometric test. (C) Analysis of enriched motifs in genomic regions showing significantly increased chromatin accessibility by the combination treatment. (D) Footprint analysis of ATAC-seq data for the combination treatment. Volcano plots indicate the transcription factor footprints that are most significantly altered following combination treatment. Transcription factors were classified alphabetically and divided among the three volcano plots. Statistical significance was assessed using Wald test, and the resulting $p$ values were adjusted for multiple testing using the Benjamini–Hochberg procedure to control the false discovery rate. (E) Correlation plot of combined ATAC-seq and RNA-seq data. Significant genes in both RNA-seq and ATAC-seq analyses were plotted. Statistical analysis was performed using the Pearson correlation test.

