## [Peer Review File · EMBO Molecular Medicine]

Dual targeting of CDK6 and LSD1 is synergistic and overcomes differentiation blockade in AML

Lise BRAULT, Edwige VOISSET, Mathieu Desaunay, Antonia BOUDET, Paraskevi Kousteridou, Sebastien Letard, Nadine Carbuccia, Armelle Goubard, Rémy Castellano, Yves Collette, Julien Vernerey, Isabelle VIGON, Jean-Max Pasquet, Patrice DUBREUIL, Sophie LOPEZ, and Paulo De Sepulveda

Corresponding author: Paulo De Sepulveda (paulo.de-sepulveda@inserm.fr)

Review Timeline:

Submission Date:	18th Oct 24
Editorial Decision:	19th Nov 24
Revision Received:	30th May 25
Editorial Decision:	9th Jul 25
Revision Received:	29th Jul 25
Accepted:	4th Aug 25

Editor: Lise Roth

Transaction Report:

19th Nov 2024

Dear Dr. De Sepulveda,

Thank you for the submission of your manuscript to EMBO Molecular Medicine. We have now heard back from the referees who agreed to evaluate your manuscript. As you will see below, the reviewers raise substantial concerns on your work, which unfortunately preclude its publication in EMBO Molecular Medicine in its current form.

Indeed, the reviewers acknowledge the translational interest of the manuscript in light of the critical need for AML therapies, but they also raise concerns related to the limited mechanistic insight and limited conclusiveness of in vivo studies.

Addressing the reviewers concerns in full will be necessary for further considering the manuscript in our journal. Revising the manuscript according to the referees' recommendations appears to require a lot of additional work and experimentation, and given the potential interest of your findings, we are ready to extend the deadline to 6 months with the understanding that acceptance of the manuscript would entail a second round of review.

EMBO Molecular Medicine encourages a single round of revision only and therefore, acceptance or rejection of the manuscript will depend on the completeness of your responses included in the next, final version of the manuscript. For this reason, and to save you from any frustrations in the end, I would strongly advise against returning an incomplete revision. Should you find that the requested revisions are not feasible within the constraints outlined here and prefer, therefore, to submit your paper elsewhere, we would welcome a message to this effect.

We require:

- 1) A .docx formatted version of the manuscript text (including legends for main figures, EV figures and tables). Please make sure that the changes are highlighted to be clearly visible.
- 2) Individual production quality figure files as .eps, .tif, .jpg (one file per figure). For guidance, download the 'Figure Guide PDF' (<https://www.embopress.org/page/journal/17574684/authorguide#figureformat>).
- 3) At EMBO Press we ask authors to provide source data for the main figures. Our source data coordinator will contact you to discuss which figure panels we would need source data for and will also provide you with helpful tips on how to upload and organize the files.
- 4) A .docx formatted letter INCLUDING the reviewers' reports and your detailed point-by-point responses to their comments. As part of the EMBO Press transparent editorial process, the point-by-point response is part of the Review Process File (RPF), which will be published alongside your paper.
- 5) A complete author checklist, which you can download from our author guidelines (<https://www.embopress.org/page/journal/17574684/authorguide#submissionofrevisions>). Please insert information in the checklist that is also reflected in the manuscript. The completed author checklist will also be part of the RPF.
- 6) All Materials and Methods need to be described in the main text using our 'Structured Methods' format. According to this format, the Methods section includes a Reagents and Tools Table (listing key reagents, experimental models, software and relevant equipment and including their sources and relevant identifiers) followed by a Methods and Protocols section describing the methods, ideally using a step-by-step protocol format. The aim is to facilitate adoption of the methodologies across labs. Please download and fill our Reagents and Tools Table template (.docx), which you can find in our author guidelines: <https://www.embopress.org/page/journal/14693178/authorguide#structuredmethods>. When submitting your revised manuscript, please do not include the Reagents and Tools Table in the Methods section of the manuscript but upload it as a separate file choosing the file type "Reagent Table". An example of a Method paper with Structured Methods can be found here: <https://www.embopress.org/doi/10.15252/msb.20178071>
- 7) It is mandatory to include a 'Data Availability' section after the Materials and Methods. Before submitting your revision, primary datasets produced in this study need to be deposited in an appropriate public database, and the accession numbers and

database listed under 'Data Availability'. Please remember to provide a reviewer password if the datasets are not yet public (see <https://www.embopress.org/page/journal/17574684/authorguide#dataavailability>).

8) For data quantification: please specify the name of the statistical test used to generate error bars and P values, the number (n) of independent experiments (specify technical or biological replicates) underlying each data point and the test used to calculate p-values in each figure legend. The figure legends should contain a basic description of n, P and the test applied. Graphs must include a description of the bars and the error bars (s.d., s.e.m.). Please provide exact p values.

12) Author contributions: CRediT has replaced the traditional author contributions section because it offers a systematic machine readable author contributions format that allows for more effective research assessment. Please remove the Authors Contributions from the manuscript and use the free text boxes beneath each contributing author's name in our system to add specific details on the author's contribution. More information is available in our guide to authors.

13) Every published paper now includes a 'Synopsis' to further enhance discoverability. Synopses are displayed on the journal webpage and are freely accessible to all readers. They include a short stand first (maximum of 300 characters, including space) as well as 2-5 one-sentences bullet points that summarizes the paper. Please write the bullet points to summarize the key NEW findings. They should be designed to be complementary to the abstract - i.e. not repeat the same text. We encourage inclusion of key acronyms and quantitative information (maximum of 30 words / bullet point). Please use the passive voice. Please attach these in a separate file or send them by email, we will incorporate them accordingly.

Please also suggest a visual abstract to illustrate your article as a PNG file 550 px wide x 300-600 px high. A cropped portion of this image will serve as thumbnail for the table of content on our webpage.

14) As part of the EMBO Publications transparent editorial process initiative (see our Editorial at <http://embomolmed.embopress.org/content/2/9/329>), EMBO Molecular Medicine will publish online a Review Process File (RPF) to accompany accepted manuscripts.

In the event of acceptance, this file will be published in conjunction with your paper and will include the anonymous referee reports, your point-by-point response and all pertinent correspondence relating to the manuscript. Let us know whether you agree with the publication of the RPF and as here, if you want to remove or not any figures from it prior to publication. Please note that the Authors checklist will be published at the end of the RPF.

EMBO Molecular Medicine has a "scooping protection" policy, whereby similar findings that are published by others during

review or revision are not a criterion for rejection. Should you decide to submit a revised version, I do ask that you get in touch after three months if you have not completed it, to update us on the status.

I look forward to receiving your revised manuscript.

Yours sincerely,

Lise Roth

***** Reviewer's comments *****

Referee #1 (Remarks for Author):

The paper entitled „ Dual targeting of CDK6 and LSD1 is synergistic and overcomes differentiation blockade in AML" by Brault et al. aims to highlight a novel combinatorial treatment strategy for AML by applying the CDK4/6 kinase inhibitor palbociclib and a LSD1 inhibitor. The authors show that the combination treatment increases myeloid differentiation and cell death of AML cell lines and primary patient samples in vitro and in vivo. This is underlined by chromatin changes when CDK6 and LSD1 are inhibited. This research has a certain significance in understanding the role of CDK6 inhibition for AML therapy and underlines a novel combinatorial treatment option, which is desperately needed.

My specific comments are as follows:

Figure 1:

- The authors show the effects of the combinatorial treatment upon pre-treatment with palbociclib. What if they don't pretreat?

Figure 2:

- Provide information on the survival of the single treated controls (palbociclib and TCP).
- Did the authors use the same cell numbers to replat the colonies in Fig. 2g?

Figure 4:

- Why did the authors not perform palbociclib pretreatment as in the in vitro assays? Would this impact the outcome?
- Please provide information on the % of leukemic cells in the blood
- It is not clear if Fig. 4d-g are data from the same experiment? Is there an explanation for the PDX#3 why it is not responding, eg. Mutation status, therapy?

Figure 5:

- What are the effects of the treatments on healthy cells? The authors should perform the replating assay with eg. CD34+ cord blood cells.
- The authors conclude effects on the cell cycle state by the combinatorial treatment. Data to show changes in proliferation and apoptosis would help to better understand response mechanisms.

Figure 6:

- It needs to be clarified if a palbociclib pretreatment was performed and if this has any impact on gene regulation.

Figure 7:

- Are the transcription factors associated with differentiation deregulated on mRNA or protein levels? Is there a link between CDK6, CEBPA and SPI1?

Figure 8:

- Data from AML samples should be combined with the data of the MV4-11 cells. What is the overlap in DEGs?

Figure 9:

- A side by side analysis of the MV4-11 cells and the AML samples regarding the apoptosis pathways should be provided.
- Is the increased cell death a follow-up of differentiated cells? Analysis over a longer time period is required starting right after the treatment start.
- Findings should be validated in primary patient samples

Minor:

- Make clear if there was a pretreatment in Fig. 1g

- Increase the replicates in Fig. 1 e
- Why is there a difference in the sample number between the graphs in Fig. 1g
- State in the text that a colony assay has been performed in Fig. 2e
- Figures could be combined to reduce figure numbers, eg. Fig 7 and 8
- The manuscript could be improved by a better description of the experiments in the text.

Referee #2 (Comments on Novelty/Model System for Author):

Work with cell lines and Xenotransplant experiments with cells lines are well performed and informative.

PDX experiments not conclusive due to variability in response making it hard to predict the translational importance of this study.

Mechanistic studies are superficial.

Referee #2 (Remarks for Author):

Brault et al., describe findings that the combined inhibition of CDK6 and LSD1 restores myeloid differentiation and depletes the leukemic progenitor compartment in AML samples. The authors use a 19-drug screen to identify that LSD1 inhibition synergizes with CDK 4/6 inhibitors. These data were confirmed in vitro using cell lines and AML patient samples. Then the authors test this sensitization in xenotransplants with MV4;11 cells and further using 5 PDX samples.

Major comments:

In vivo PDX treatments. The authors need to provide more data specific for each experiment. What was the percentage of human blasts in PB of mice when dosing was initiated? The authors state "In these models, 4 out of 5 PDX mice treated with the combination of CDK6 and LSD1 inhibitors showed a strong reduction in leukemic cells in both bone marrow and spleen (Fig. 4d-e)." The data in figure 4 d-e seems to be best interpreted as 2 PDX samples (#2, 4) had a good response of >3 fold reduction of human blasts, 1 sample (#3) had no response, and 2 samples (#1,5) had a modest response of 10-30% reduction. While it might be statistically significant these responses are not dramatic. Also, the authors describe dosing of mice as 4 days on followed by 3 days off. What is the rationale for this type of dosing? Why did the authors not use conventional continuous dosing for all days in the experiment? Was this due to toxicity? It would be useful if the authors collected blood samples followed by FACS analyses for human cells during the treatment of PDX mice to assess kinetics of the drug responses.

PDX#2 has an MLL1 translocation and showed good response. Also 3 of 6 tested cells lines carried MLL1 translocations. Maybe the authors should have selected this genotype of PDX for their experiments instead of using a mix of AML subtypes?

The genetic mutations of PDXs 3, 4, and 5 are somewhat similar (NPM1, TET2, DNMT3A, "TyK-R"+) according to sup table 1. However, the responses of these PDX to treatment varied dramatically. How would the authors interpret this difference in response and how would one consider using these findings to guide clinical translation?

Gene expression and chromatin accessibility analyzes are underdeveloped. Identifying gene expression signatures that are either up or down after treatment is not terribly informative in and of itself. The authors state in the introduction "Mechanistically, this combination induces major changes in chromatin accessibility " - however in the main text they do not provide details what these changes mean. It would be very helpful if they could connect chromatin accessibility changes with gene expression changes as a way to begin to understand some of the mechanisms. Significant work in this section of the manuscript is needed to understand what these inhibitors are doing to leukemia cells.

Finally, most of the in vitro experiments are performed with Tranylcypromine which is least potent and selective LSD1 inhibitor. At least some of the experiments should be validated with better, clinically relevant, LSD1 inhibitors.

Minor comments:

Figure 1b - please label the cell lines in bar graphs like in figure 1c.

Referee #3 (Comments on Novelty/Model System for Author):

technical quality: mostly well done but the authors may lack some technical knowledge with some of the in vitro assays used.
Novelty: is medium as the inhibition of CDK6 or LSD1 has already been reported to treat AML, only with exception when combined.

Medical impact: is always high as despite many years of research this disease is still mostly incurable with high rates of relapse and dismal prognosis (especially for patients aged >60) hence any potential treatment or study about the biology of AML is very important.

model system: mostly adequate, but some assays were not. please refer below for specifics.

Referee #3 (Remarks for Author):

In this work, Brault et al tested whether combining the inhibition of LSD1 and CDK6 could overcome the differentiation blockage of acute myeloid leukemia (AML). The authors used AML cell lines, primary cells, in vitro and in vivo assays, and molecular studies to support their hypothesis that the combined treatment is more effective than the respective single inhibition. In general, is a well-done work and the data presented is mostly supportive. However, some clarifications and corrections are required to improve the quality and the messaging of this manuscript.

Major concerns:

-the manuscript is too long with too many Figures, some of them with similar messaging and others less important to be as the main figure. Hence, some data should be moved to supplementary and/or deleted and be more focused on the messaging. Such as Figures 2 and 5 can be combined and be more concise (also the respective text) in terms of the impact of the clonogenic potential upon the different treatments using 1 or 2 AML cell lines as models throughout. Figure 5a can be made as one simple quantification graph instead of the flow histograms. Data shown in Figure 3 although interesting but has less weight as compared to the rest to be as main figure. Some of the supplementary figures can be combined to make them less numerous.

-Figure 2d: transplantation of the AML cell line or any cell line after in vitro treatment is not the best experiment to demonstrate the impact of drug testing in vivo (which is the most relevant than any in vitro assay), even when cells are transplanted using same number of cells in all the testing conditions. The authors showed mice survival without the leukemic burden in these mice, hence it was difficult to evaluate the contribution of this data. The author showed properly the in vivo data in Figure 4a and 4b. As such, should delete this data as serves no purpose to their messaging.

-Figure 2e: although colony assay is a common assay used to assess progenitor activity in normal HSPCs but also can be used in AML cell lines. However, this assay is well-known for being not very reliable for primary AMLs. As a reference please consult this paper as an example PMID: 31936647. Because many AML cells don't form colonies and when they do, many come from residual non-leukemic cells. hence, it makes it difficult to know whether the colonies detected were of leukemic origin or not and consequently the interpretation of the results. Although some publications reported its use, but it does not mean they were done properly and the interpretation of these published data should be done with caution. Either the authors validate that the colonies detected were leukemic (with their respective genetic lesion(s)) or the authors should delete all CFU data using primary AML cells. Alternatively, move the data to a supplementary Figure and comment on the potential pitfalls of this assay.

-one of the major deficits of this work is the lack of comparison of using cytarabine (ARAC)+ an anthracycline treatment in vitro and in vivo with the combination of LSD1+CDK6 inhibition to show how effective the latter combination is as compared to the traditional regime. More importantly, the authors conveyed a few times (in the introduction and then in the discussion) the potential effective use of this new LSD1+CDK6 inhibition combination to treat AMLs that are resistant to the conventional regime. However, the experiments were not designed to tackle this important point. E.g. were all the primary samples used, especially samples 1 to 5 used in the PDX experiments resistant to the conventional regime and hence were selected for the in vivo studies? Tone down the messaging or provide experiments to support this point.

-the interpretation of the molecular data: the part that is related to the "dual CDK6/LSD1 targeting restores the expected transcriptional response associated with LSD1 inhibition", is interesting but the authors did not further explore/validate with experiments. It was only based on the transcriptomic data. The authors should make this part more concise. Instead, they should explore how their molecular data supports the whole in vitro data in particular the clonogenic and differentiation impact of the combination treatment which data occupied most of this manuscript.

EMBO Molecular Medicine- EMM-2024-20757

Brault et al, Dual targeting of CDK6 and LSD1 is synergistic and overcomes differentiation blockade in AML.

RESPONSES TO THE REVIEWERS' COMMENTS

We would like to thank the editor and reviewers for their time and constructive feedback on our manuscript. Their insightful comments have been instrumental in improving the clarity of the text, refining the figures, and strengthening our conclusions.

Our detailed responses to each comment are provided below in blue.

We hope that our revisions meet the expectations of the reviewers and that the manuscript is now suitable for publication in *EMBO Molecular Medicine*.

Please note that we changed the previous Figure 6b (now Fig. 5b) because the thresholds used for the figures differed between panel 6a ($p < 0.01$) and panel 6b ($p < 0.05$). Both now correspond to a statistical threshold of $p < 0.01$. The correction does not change the interpretation/conclusions.

Referee #1 (Remarks for Author):

The paper entitled „ Dual targeting of CDK6 and LSD1 is synergistic and overcomes differentiation blockade in AML" by Brault et al. aims to highlight a novel combinatorial treatment strategy for AML by applying the CDK4/6 kinase inhibitor palbociclib and a LSD1 inhibitor. The authors showed that the combination treatment increases myeloid differentiation and cell death of AML cell lines and primary patient samples in vitro and in vivo. This is underline by chromatin changes when CDK6 and LSD1 are inhibited. This research has a certain significance in understanding the role of CDK6 inhibition for AML therapy and underlines a novel combinatorial treatment option, which is desperately needed.

My specific comments are as follows:

Figure1:

- *The authors show the effects of the combinatorial treatment upon pre-treatment with palbociclib. What if they don't pretreat?*

We opted for a pretreatment in our in vitro studies based on preliminary data (Figure below) showing that pre-exposure to palbociclib, (green) led to slightly greater efficacy compared to no pretreatment. Importantly, the combination also remains effective without pretreatment (see red vs green in the figure below), though to a slightly lesser extent. Other combinations were also explored, as illustrated in panels A-C (figure below).

A. Schedule of the experiment (attention must be paid to the color code);
B. Increased expression of the CD11b differentiation marker relative to the mock-treated cells (black bar);
C. Decreased number of colony-forming cells compared to the mock-treated cells (black bar).

Figure 2:

- Provide information on the survival of the single treated controls (palbociclib and TCP).

The purpose of the experiment was to correlate the results of colony-forming assays with in vivo leukemia reconstitution. In our initial experiments, we did not include the single TCP treatment control due to its lack of effect on colony formation.

We have now repeated the experiment, including all single treatments alongside the combination treatment. As anticipated, TCP-treated cells behaved similarly to mock-treated control cells. Both palbociclib and the combination treatment extended mouse survival, consistent with the observed reduced CFC potential. It should be noted that mice treated with palbociclib alone showed a slightly shorter (though not statistically significant) survival compared to the combination group.

We moved the previous Figure 2 panel d together with the new experiment in **Appendix Figure S1c** (as requested by the Reviewer #3).

- Did the authors use the same cell numbers to replat the colonies in Fig. 2g?

Yes, secondary and tertiary methylcellulose assays were always performed by plating an equal number of cells from the previous plates. The information is provided in the Methods section, under the "Colony-forming cell assays" paragraph.

Figure 4:

- Why did the authors not perform palbociclib pretreatment as in the in vitro assays? Would this impact the outcome?

As mentioned above, the effects in vitro were similar without pretreatment, with only a slight improvement observed following a 4-day treatment. Because the in vivo treatments lasted for 4 weeks, we decided to simplify the protocol and did not include a pretreatment. Given the positive outcome observed, we did not compare this protocol with a pretreatment approach in order to avoid an unnecessary increase in animal use.

- Please provide information on the % of leukemic cells in the blood

A new panel has been added to **Expanded View Fig.4** with the requested information.

- It is not clear if Fig. 4d-g are data from the same experiment?

Figure 4d-e (testing 5 different PDX mice – now presented as **Figure 3d** and **Expanded View 4a**) and Figure 4f-g (in vivo results comparing single vs combination treatments on 2 PDX mouse models – now **Figure 3e-f**) are from independent experiments. This information has been included to the figure legend.

- Is there an explanation for the PDX#3 why it is not responding, eg. Mutation status, therapy?

We have currently no clear explanation for the lack of response observed in PDX#3. This sample shares mutations with other PDX models used in our study (see Expanded View Table 1, samples 1-5); yet, as expected with primary AML patient samples, each model remains unique. Heterogeneous responses among unrelated primary samples are common and expected, and reflect the variability seen in clinical settings, where patients within the same genetic subgroup respond differently to therapy.

The lack of response in this particular sample is certainly noteworthy. Interestingly, PDX#3 was responsive in vitro, suggesting that in vivo resistance may be driven by interactions with the in vivo microenvironment that provide protection. This raises the possibility that PDX#3 represents a model of non-cell-autonomous resistance, in which protective signals from the microenvironment contribute to therapeutic failure.

Dissecting such resistant mechanisms is complex and would require a dedicated, in-depth study of its own.

Figure 5:

- What are the effects of the treatments on healthy cells? The authors should perform the replating assay with eg. CD34+ cord blood cells.

Both CDK6 and LSD1 inhibition have been reported to impact normal CD34+ blood cells in vitro. Furthermore, normal progenitor cells could not be maintained in culture long enough to perform the requested experiments.

In response to the reviewer's comment, we attempted replating assays using human CD34+ cells. As illustrated below, under our culture conditions and timing, normal progenitor cells cannot be maintained, with a marked decrease already evident at the first replating on methylcellulose (2nd plating; see control cells -black bar- in the histograms). Consequently, the in vitro replating assay is not informative in this specific context, as the low number of control progenitors precludes a reliable assessment of treatment effects.

Therefore, the most relevant data on the normal hematopoietic compartment are those obtained in vivo from treated-mice (Supplementary Figure 6 of the original submission – now **Expanded View 4**).

- The authors conclude effects on the cell cycle state by the combinatorial treatment. Data to show changes in proliferation and apoptosis would help to better understand response mechanisms.

Cell cycle arrest occurs upon inhibition of CDK4/6 by palbociclib. In Figure 5 (now **Figure 4**), we show that the combination treatment has long-term effects on the cell cycle, even after drug removal, unlike palbociclib alone.

Regarding proliferation changes, palbociclib is a well-described reversible inhibitor, which is illustrated in the figure below: although palbociclib-treated cells (blue line) require time to re-enter the cell cycle and resume proliferation, they do expand once palbociclib is removed (time 0 in the Figure below). In contrast, cells treated with the combination (green line) fail to recover, indicating a more sustained anti-proliferative effect.

Regarding cell death, the results are illustrated in Figure 9 of the original manuscript (now **Figure 7**).

Figure 6:

- It needs to be clarified if a palbociclib pretreatment was performed and if this has any impact on gene regulation.

As indicated in the Methods section, all our in vitro experiments followed the same “1+3” treatment scheme: 1 day of palbociclib alone, followed by 3 days of combination treatment with palbociclib and an LSD1 inhibitor.

Figure 7:

- Are the transcription factors associated with differentiation deregulated on mRNA or protein levels? Is there a link between CDK6, CEBPA and SPI1?

We are not aware of a direct link between CDK6 and CEBP or SPI1. However, LSD1 is well known to indirectly regulate SPI1-binding to gene regulatory sequences through its interaction with GFI1.

At the mRNA level, several primary AML samples showed changes in gene expression following combination treatment, although others did not, which is expected regarding the heterogeneity of AML. About CEBPA and SPI1 protein levels, we did not observe notable changes. Nevertheless, our preliminary results suggest that SPI1 and CEBPB may be upregulated at the mRNA expression level following treatment, but further investigation using a larger cohort will be necessary to conclude on this question.

Below are the mRNA expression analyses (TPM) of 4 relevant myeloid transcription factors in 6 primary AML samples, and the impact of the combination treatment (Combo) on their expression:

Ctrl= mocked-treated cells; Combo= cells treated with the combination of CDK6 and LSD1 inhibitor (for a total of 4 days, consistent with all in vitro experiments in our study).

Figure 8:

- Data from AML samples should be combined with the data of the MV4-11 cells. What is the overlap in DEGs?

We overlapped the MV4-11 and AML sample data to extract the common DEGs. The data is displayed as **Appendix Figure S8b**. Note that 523 DEGs are common, and 53% of the DEGs found in patient samples are also significantly deregulated in MV4-11.

In addition, we also performed analyses of the overlap between RNA-seq and ATAC-seq data in response to comments from Reviewers #2 and #3. Remarkably, DEG expression levels correlated with chromatin accessibility (Expanded View Figure 5d).

Figure 9:

- A side by side analysis of the MV4-11 cells and the AML samples regarding the apoptosis pathways should be provided.

- Findings should be validated in primary patient samples

The original Figure 9 indicated increased cell death (9a) and sub-G1 cells (9b), and evidence supporting ferroptosis as the main cell death pathway (9c and 9d) involved in two AML cell lines, reinforced by the increased lipid oxidation (9d - now **Figure 7**).

The analysis of cell death in primary AML patient samples present a real challenge, as unlike cell line models, these primary cells exhibit high levels of cell death in culture. Therefore, complex experiments done on cell lines cannot be done on primary samples.

Nevertheless, in order to strengthen our data, we now provide additional data increased cell death in combination-treated primary patient samples. Furthermore, we now include a BODIPY staining analysis of two primary samples showing increased lipid oxidation upon treatment (below and **Appendix Figure S10d-e**).

(Left) Quantification of live cells (DAPI negative) in control vs treated cells (n=16).

(Right) Quantification of lipid oxidation in 2 primary AML samples using C11-BODIPY 581/591.

- Is the increased cell death a follow-up of differentiated cells? Analysis over a longer time period is required starting right after the treatment start.

In our experiments, the cells do not reach terminal differentiation (macrophage stage or fully differentiated granulocytes) in the 4-day protocol; it is then unlikely that the observed death is a consequence of cell differentiation. Furthermore, in our experiments, both cell death and differentiation increase over time (Figure below for cell death). We therefore suggest that both phenomena occur simultaneously.

Minor:

- Make clear if there was a pretreatment in Fig. 1g

All our in vitro experiments were done on the same "1+3" treatment scheme: 1 day of palbociclib alone, followed by 3 days of combination treatment with palbociclib and an LSD1 inhibitor.

- Increase the replicates in Fig. 1 e

A third independent experiment has been performed, which is now included in the figure.

- Why is there a difference in the sample number between the graphs in Fig. 1g

While CD11b (early myeloid marker) was monitored from the beginning, CD86 (later-stage myeloid marker) was not evaluated in our first set of experiment with primary AML cells, thus explaining the 3 additional primary samples analyzed for CD11b.

- State in the text that a colony assay has been performed in Fig. 2e

We have now amended the text as suggested: "We then evaluated these effects on leukemic progenitors on 24 adult primary AML patient samples, using methylcellulose colony assays".

- Figures could be combined to reduce figure numbers, eg. Fig 7 and 8

We reduced the number of Figures to 7. There are 5 Expanded view Figures, and all other Figures are now part of Appendix Figures.

Original Figures 7 and 8 represent different types of data (ATAC-seq analyses of MV4-11 cells vs RNA-seq analyses of primary AML samples). Nonetheless, to reduce the total number of figures, we combined the original Figures 6 and 7 (RNA-seq and ATAC-seq analyses on MV4-11 cell line – now **Figure 6**). We also transferred the original Figure 3 and several figure panels as Expanded View Figures, as recommended by other reviewers.

- The manuscript could be improved by a better description of the experiments in the text.

We have made several amendments to the text to improve clarity and hope that Reviewer #1 will find them satisfactory.

Referee #2 (Remarks for Author):

Brault et al., describe findings that the combined inhibition of CDK6 and LSD1 restores myeloid differentiation and depletes the leukemic progenitor compartment in AML samples. The authors use a 19-drug screen to identify that LSD1 inhibition synergizes with CDK 4/6 inhibitors. These data were confirmed in vitro using cell lines and AML patient samples. Then the authors test this sensitization in xenotransplants with MV4;11 cells and further using 5 PDX samples.

Major comments:

In vivo PDX treatments. The authors need to provide more data specific for each experiment. What was the percentage of human blasts in PB of mice when dosing was initiated?

Engraftments were monitored weekly by detecting circulating hCD45⁺/hCD33⁺ blasts in peripheral blood. Treatments were initiated upon detection of blasts in the bloodstream (usually a few blasts per microliter of blood).

The sentence in the Methods section originally was: "Once hCD45⁺/hCD33⁺ blasts were detected in the bloodstream, mice were allocated to treatment groups ...". We have now added "and the treatments started" to the end of the sentence to clarify that the treatment began immediately upon detection of blasts.

The Table below provides more data on each PDX experiment:

	PDX#1	PDX#2	PDX#3	PDX#4	PDX#5
Start date of the treatment (days after injection)	42	18	17	31	24
Duration of the treatment (days)	31	24	18	18	25

The authors state "In these models, 4 out of 5 PDX mice treated with the combination of CDK6 and LSD1 inhibitors showed a strong reduction in leukemic cells in both bone marrow and spleen (Fig. 4d-e)." The data in figure 4 d-e seems to be best interpreted as 2 PDX samples (#2, 4) had a good response of >3 fold reduction of human blasts, 1 sample (#3) had no response, and 2 samples (#1,5) had a modest response of 10-30% reduction. While it might be statistically significant these responses are not dramatic.

We acknowledge that the word "strong" was inappropriate and removed it from the text as it does not accurately describe the data.

Regarding the reviewer's comment on the "modest response", we would like to emphasize 4 points:

1. Variability in AML mouse models is expected; it is inherent to both the heterogeneity of primary samples and technical variability related to cell transplantation and recurrent injections (treatments).

2. The variability in treatment response is consistent with the clinical reality of AML, where even within specific subgroups, patient responses can vary widely, regardless of the drug used.
3. Despite the fact that the treatment was suboptimal, we observed a significant reduction in leukemic burden in 4 out of 5 of PDX mouse models.
4. We would like to draw attention to reference studies published in prestigious journals, in which comparable or even more modest effects were reported. For example, the figure below shows the reduction observed in the published manuscript that led to clinical trials of iadademstat (ORY-1001) in AML (*Maes et al, Cancer Cell, 2018*). Please note the overlap between control and treated animals, and the simpler model used (subcutaneous injection of an AML cell line) compared to our study, which involved primary human AML samples and analysis of the leukemia across various hematopoietic sites.

(C) Mean tumor volume of a subcutaneous MV(4;11) xenograft model on day 33 treated as in (B). Each dot represents one mouse (n = 10 per treatment group). p value 0.02 mg/kg ORY-1001 versus vehicle = 0.0047; p value 0.03/0.02 mg/kg ORY-1001 versus vehicle = 0.0133 (one-way ANOVA and Dunnett's multiple comparison test).

In conclusion, we agree to moderate the strength of our conclusion, but we respectfully disagree with the initial statement that "*PDX experiments not conclusive due to variability in response making it hard to predict the translational importance of this study.*" Instead, we argue that we observed a reduction of leukemic burden in 4 out of 5 highly heterogeneous PDX models, despite suboptimal treatment conditions (as further discussed in the following comment).

Also, the authors describe dosing of mice as 4 days on followed by 3 days off. What is the rationale for this type of dosing? Why did the authors not use conventional continuous dosing for all days in the experiment? Was this due to toxicity?

As we indicated in the discussion, "the treatment was discontinued due to the extreme sensitivity of the transplanted NSG mouse models". Indeed, compared to the doses used in some reference publications, we had to lower the concentration of palbociclib to 25 mg/kg, and even then, we could not treat mice for more than 4 consecutive days. With the help of the local ethics committee, we set up the experimental schedule described in the manuscript to increase the tolerability. Since we obtained positive results and to comply with ethical guidelines related to the number of mice used (not mentioning the cost of an NSG mouse cohort), we did not pursue further optimization schedules.

We argue that despite these non-ideal conditions (i.e. lack of continuous treatment), we still observed a convincing decrease in leukemic blasts in 1 cell line-derived xenograft mouse model, and in 4 out of 5 AML-PDX models.

It would be useful if the authors collected blood samples followed by FACS analyses for human cells during the treatment of PDX mice to assess kinetics of the drug responses. The analysis of blood samples collected from the PDX mouse experiments had already been performed, and the data are now included in **Extended View Figure 4b**.

PDX#2 has an MLL1 translocation and showed good response. Also 3 of 6 tested cells lines carried MLL1 translocations. Maybe the authors should have selected this genotype of PDX for their experiments instead of using a mix of AML subtypes?

We are confident that our data are relevant for most AML subtypes. Therefore, by focusing only on MLL translocations at this stage, we would narrow the general message of the study. Nevertheless, we agree with the reviewer that MLL translocations could represent a subtype of interest for future translational studies, and we have added two sentences in the Discussion to highlight this point: "Several AML models used in the study carried MLL-rearrangements, including PDX#2 which was the best responder in vivo. Given the requirement for novel therapies in this category with poor prognosis, further preclinical studies on AML with MLL-translocations might warrant further investigation."

The genetic mutations of PDXs 3, 4, and 5 are somewhat similar (NPM1, TET2, DNMT3A, "TyK-R"+) according to sup table 1. However, the responses of these PDX to treatment varied dramatically. How would the authors interpret this difference in response and how would one consider using these findings to guide clinical translation?

Samples appear to carry similar mutations however, they differ significantly in various aspects, including the specific mutations present, variant allele frequencies (VAF), clonal composition and state of differentiation, emphasizing that each patient-derived AML sample is unique. For instance, PDXs #3, #4 and #5 are classified as M2, M4, and M1, respectively, meaning distinct stages of myeloid differentiation. Although they share some mutations, the overall mutational profiles are different, and notably, PDX#5 lacks a TK mutation. Finally, PDX#3 and #4 exhibit normal karyotypes, whereas PDX#5 has a complex karyotype.

Our findings are still at an early stage to guide clinical translation. At this stage, we have demonstrated (1) that a range of samples respond to the combination, which is an encouraging feature at least hypothetically, to limit leukemia cell plasticity and resistance. (2) Moreover, we showed that repositioning existing CDK4/6 inhibitors can improve the efficacy of existing drugs (LSD1 inhibitors); (3) this combination could be an option when cells exhibit resistance to conventional therapy; Our study included four cytarabine-resistant cell line models, four PDX samples resistant to the FLT3 inhibitor Quizartinib, and several patient samples obtained at relapse, a stage typically characterized by chemotherapy resistance. (see our answers to reviewer #3); and (4), because the combination shows synergy, these molecules could be used at lower concentrations, potentially leading to fewer secondary effects.

Gene expression and chromatin accessibility analyzes are underdeveloped. Identifying gene expression signatures that are either up or down after treatment is not terribly informative in and of itself. The authors state in the introduction "Mechanistically, this combination induces major changes in chromatin accessibility" - however in the main text they do not provide details what these changes mean. It would be very helpful if they could connect chromatin accessibility changes with gene expression changes as a way to begin to understand some of the mechanisms. Significant work in this section of the manuscript is needed to understand what these inhibitors are doing to leukemia cells.

We have added further analyses to this part of the manuscript.

- We have validated some gene expression data (part of the LSD1 signature), through quantitative PCR on independent RNA preparations, and on 3 cell surface protein using flow cytometry (shown in **Appendix Figure S6**)
- By combining the RNAseq data from the cell line model with RNAseq data of primary samples (request from Reviewer #1), we have narrowed and improve the robustness of the gene list deregulated by the combination (**Appendix S8b**).
- -By studying footprint analyses, we show that CEBP and SPI1 (two main differentiation factors) are among the transcription factors that gain binding to regulatory genomic regions (**EV Figure 5c**).
- Importantly, we analyzed the correlation of RNAseq DEGs with the ATACseq data, and show a correlation (**EV Figure 5d**), mainly supported by the upregulated genes (which show increased accessibility of their regulatory regions). In other words, chromatin accessibility is very likely responsible for the increased expression of DEGs. This is a major addition, as it connects chromatin accessibility with gene regulation, differentiation programs and the observed cellular data. We hope Reviewer #2 agrees that this is a beginning of mechanistic understanding or hypothesis.

We have also added a sentence in the discussion to link the molecular observations with the cellular effects observed.

*Finally, most of the in vitro experiments are performed with Tranylcpromine which is least potent and selective LSD1 inhibitor. **At least some of the experiments should be validated with better, clinically relevant, LSD1 inhibitors.***

Most of the in vitro experiments presented in the manuscript were indeed performed using TCP, as it was the original hit identified in our screen of small molecules. However, whenever we changed to another LSD1 inhibitor (mainly ORY-1001, also known as iadademstat), **we consistently observed comparable results than with TCP.**

Moreover, the main message of the original Figure 3 (now **Expanded View Figures 2 and 3**) was to demonstrate that LSD1 is indeed the target of TCP and other LSD1 inhibitors. To this end, we used two other LSD1 inhibitors as well as RNA interference to reduce LSD1 expression.

Regarding clinically relevant LSD1 inhibitors, the most advanced LSD1 inhibitor for AML is **ORY-1001** (iadademstat; PMID: 33052756). In the original version of the manuscript, ORY-1001 was used in:

- **Figure 3c** (CD11b expression in MV4-11 cells treated with ORY-1001 alone or in combination with palbociclib, ribociclib, or abemaciclib),
 - **Figure 3e** (CD11b expression in MV4-11 cells treated with ORY-1001 in combination with CDK6-targeting RNA interference),
 - **Supplementary Figure 4** (now **Expanded View Figures 2** - colony formation assay).
- In addition, **GSK2879552** (used recently by *Ciceri et al, Nature, 2024*; doi.org/10.1038/s41586-023-06984-8) was also evaluated in **Figure 3c**, either alone or in combination with each of the three CDK6 inhibitors.
- Since our responses to the Reviewers will be available alongside the manuscript, we have also included below several other experiments showing that ORY-1001 demonstrates similar activity to TCP in vitro, using various AML cell lines and primary samples.

Comparison of ORY-1001 and TCP in combination with palbociclib on colony forming assays on 2 primary patient samples:

Comparison of ORY-1001 and TCP in combination with palbociclib on myeloid markers (3 primary patient samples):

Comparison of ORY and TCP in combination with palbociclib on MV4-11 cell death (DAPI positive cells):

Experiments with ORY-1001 that recapitulate results shown in the manuscript with TCP:

Minor comment:

Figure 1b - please label the cell lines in bar graphs like in figure 1c. The modification has been made as requested.

Referee #3 (Remarks for Author):

In this work, Brault et al tested whether combining the inhibition of LSD1 and CDK6 could overcome the differentiation blockage of acute myeloid leukemia (AML). The authors used AML cell lines, primary cells, in vitro and in vivo assays, and molecular studies to support their hypothesis that the combined treatment is more effective than the respective single inhibition. In general, is a well-done work and the data presented is mostly supportive. However, some clarifications and corrections are required to improve the quality and the messaging of this manuscript.

Major concerns:

-the manuscript is too long with too many Figures, some of them with similar messaging and others less important to be as the main figure. Hence, some data should be moved to supplementary and/or deleted and be more focused on the messaging. Such as Figures 2 and 5 can be combined and be more concise (also the respective text) in terms of the impact of the clonogenic potential upon the different treatments using 1 or 2 AML cell lines as models throughout. Figure 5a can be made as one simple quantification graph instead of the flow histograms. Data shown in Figure 3 although interesting but has less weight as compared to the rest to be as main figure. Some of the supplementary figures can be combined to make them less numerous.

While we may not fully agree with all aspects of the comment, we acknowledge that some sections of the manuscript and some Figures could benefit from streamlining to reduce redundancy. We are convinced, however, that including a broad range of AML cell lines (some with very different characteristics) is necessary to demonstrate that the observed results are not limited to a specific cell model.

We have reorganized the figures to address all three reviewers' comments and incorporate new experimental data. Some figures, such as the previous Figure 3, have been moved to supplementary Expanded View Figures, while others are now in the Appendix Figures. In the main Figures, RNAseq and ATAC seq data from the cell line were joined in one Figure.

The main text was modified to integrate the data from primary samples with the cell line data. This allowed us to remove redundant text. For example, the description of the RNA-seq data on primary samples was reduced by half. We thank the reviewer for helping us improve the manuscript.

-Figure 2d: transplantation of the AML cell line or any cell line after in vitro treatment is not the best experiment to demonstrate the impact of drug testing in vivo (which is the most relevant than any in vitro assay), even when cells are transplanted using same number of cells in all the testing conditions. The authors showed mice survival without the leukemic burden in these mice, hence it was difficult to evaluate the contribution of this data. The author showed properly the in vivo data in Figure 4a and 4b. As such, should delete this data as serves no purpose to their messaging.

We would like to emphasize that the goal of this experiment was to correlate the reduction in colony-forming units with the cells capacity to reconstitute leukemia, rather

than to demonstrate the in vivo impact of drug testing (which was the purpose of the original Figure 4, now **Figure 3**). The original Figure 2d complemented the in vitro analysis by supporting the colony-forming assay as a predictor of leukemia development following transplantation.

Moreover, Reviewer #1 requested to add the analysis on single-molecule treatment in Figure 2d. To accommodate both reviewers, we repeated the experiment to include these conditions, and data are now presented in **Appendix Figure S1c**.

-Figure 2e: although colony assay is a common assay used to assess progenitor activity in normal HSPCs but also can be used in AML cell lines. However, this assay is well-known for being not very reliable for primary AMLs. As a reference please consult this paper as an example PMID: 31936647. Because many AML cells don't form colonies and when they do, many come from residual non-leukemic cells. hence, it makes it difficult to know whether the colonies detected were of leukemic origin or not and consequently the interpretation of the results. Although some publications reported its use, but it does not mean they were done properly and the interpretation of these published data should be done with caution. Either the authors validate that the colonies detected were leukemic (with their respective genetic lesion(s)) or the authors should delete all CFU data using primary AML cells. Alternatively, move the data to a supplementary Figure and comment on the potential pitfalls of this assay.

We thank the Reviewer for drawing our attention to possible pitfalls, and indeed, the colony-forming assay is known to be technically challenging and requires expertise. In rare cases, we did not obtain colonies from primary AML samples, but in most cases, we were able to.

There are 3 strong arguments supporting the conclusion that colonies obtained are from leukemic progenitors and not residual non-leukemic cells:

1- We obtained colonies from PDX primary samples that had undergone 3 successive transplantations in NSG mice. With this procedure the residual non-leukemic cells of human origin are eliminated. Mouse cells are also discarded by cell sorting using human and mouse CD45 markers.

2- In our study, treatments had little to no effect on colony formation from human normal CD34+ cells, whereas **all** primary AML samples used in the study showed significant reduction. Furthermore, all primary AML samples used in our study harbored a high percentage of blasts (>70%), meaning that the residual normal cells were present at low number, making it unlikely that our results are affected by the pitfall raised here.

3- Numerous publications from various laboratories in this field validate the use and reliability of colony formation assays for characterizing leukemic progenitors in primary AML samples. As an illustration, the publication PMID: 27534895 (Nature Communications, 2016) reports that the colonies contain the genetic mutations identified in the primary samples; this was demonstrated across a large number of AML samples. Moreover, Dr. Mike Bhatia's lab at McMaster University, a prominent expert in hematology, stem cells, and leukemia, recently published a seminal study (Reference: PMID: 37433297). This research demonstrates that leukemic progenitor colony formation assays offer a robust functional measure of patients' leukemias, enabling the quantitative and rapid assessment of a wide range of patients. Here is a quote of their study: "**LPCs**

not only capture patient-specific mutations but also retain serial re-plating ability, demonstrating their biological relevance.”. Their study reports that in vitro colony formation assays, by revealing leukemic progenitor cells (LPCs), are a stronger predictor of patients' overall and event-free survival than the presence of leukemic stem cells (LSCs) detected through in vivo xenograft transplantations. Thus, LPCs provide a rapid and robust functional measure of AML disease.

Again, the publications in the field support our conclusions (and some of these show the presence of mutations in the colonies), as examples (DOI): 10.1038/s41375-021-01295-1; 10.1038/s41375-024-02312-9; 10.1038/s41586-025-08915-1; 10.1182/bloodadvances.2024013590; doi.org/10.1016/j.drug.2020.100730; 10.1038/s41375-023-02131-4; 10.1158/0008-5472.CAN-24-0577; 10.1084/jem.20200924; 10.1038/s41375-023-01835-x; 10.1038/s41375-021-01295-1; ...

Regarding the reference PMID: 31936647 quoted by Reviewer #3, although the study is of interest, the results reported have yet to be validated by other laboratories. Importantly, this short communication that stands in opposition to the conclusions of numerous other studies, does not include a Methods section which limits the possibility of comparing it directly with our work. This is particularly relevant given that the number, size and morphology of colonies obtained in colony formation assay depend on many sensitive factors, including the culture medium and FBS used, and the protocols used for sample handling and cryopreservation.

Finally, while we may disagree with the Reviewer, it is crucial to recall that several of the publications cited above convincingly demonstrate that colony assays measure the aggressivity of the AML sample. Therefore, the assay is a valuable tool to characterize primary samples.

-one of the major deficits of this work is the lack of comparison of using cytarabine (ARAC)+ an anthracycline treatment in vitro and in vivo with the combination of LSD1+CDK6 inhibition to show how effective the latter combination is as compared to the traditional regime.

Cytarabine combined with anthracycline remains the standard therapy, despite the recent introduction of novel targeted therapies (ex: FLT3 and IDH inhibitors, venetoclax). We are convinced that targeted therapies are not meant to replace conventional therapy, but rather to expand therapeutic options and address unmet clinical needs.

Our study does not aim to compare CDK6+LSD1 inhibition with the current standard of care. Instead, we explore alternative therapeutic targets and drug repositioning strategies to fulfil the need for novel therapies, particularly in cases where the standard of care is inefficient, and faces resistance. Therefore, from a clinic perspective, LSD1+CDK6 inhibition offers valuable alternative option in such resistant cases.

The interest of the combination therapy investigated in our manuscript, in regards to the conventional cytarabine/anthracycline treatment, is further discussed in the next point.

More importantly, the authors conveyed a few times (in the introduction and then in the discussion) the potential effective use of this new LSD1+CDK6 inhibition combination to treat AMLs that are resistant to the conventional regime. However, the experiments were

not designed to tackle this important point. E.g. were all the primary samples used, especially samples 1 to 5 used in the PDX experiments resistant to the conventional regime and hence were selected for the in vivo studies? Tone down the messaging or provide experiments to support this point.

We thank Reviewer #3 for highlighting this point, giving us the opportunity to clarify a major strength supporting the repositioning of CDK4/CDK6 inhibitors in AML.

There are 2 important advantages to CDK4/6 inhibitors: **(1)** all AML samples tested in our study were sensitive to palbociclib or ribociclib, regardless of their genetic alterations and classification; and **(2)** some of the samples used in our study were resistant to cytarabine and/or FLT3 inhibitors, yet remained sensitive to CDK6 inhibition. The figure below illustrates that 4 of the cell lines used in our study are resistant to cytarabine.

(1) MV4-11, THP1, SKM1 and MO7e are resistant to cytarabine (count of cells treated with 1µM cytarabine, relative to the control cells):

Furthermore, we analyzed 33 primary AML samples in total; including 6 relapse samples. Relapse samples are typically resistant to conventional chemotherapy, which is a major clinical issue, as therapeutic options are then largely limited and inefficient. Please note also that 4 out of the 5 samples used for the in vivo PDX studies are relapse samples (indicated in **Expanded View Table 1**).

Additionally, we provide below data that further support our claims. Notably, in contrast to our combination treatments, exposure to cytarabine+daunorubicin combination treatment did not impact colony-forming cell potential.

(2) In vitro cytarabine sensitivity of primary samples at diagnosis (D) and at relapse (R), showing that relapse samples showed some resistance to cytarabine.

(3) Combination treatment with cytarabine (AraC at 0.5 μ M) and daunorubicin (DNR at 10 nM) reduces cell proliferation (a), but does not impact colony-forming cells (b), and surviving cells retain leukemic activity (c):

(4) Some PDX samples (n=4 in the Figure below) used in our study are resistant to a FLT3 inhibitor but sensitive to palbociclib. Please note that in this assay, the treatment was added directly to the methylcellulose, rather than evaluating the residual colony-forming capacity after treatment:

These data were not included in the manuscript to avoid overloading it.

The interpretation of the molecular data: the part that is related to the "dual CDK6/LSD1 targeting restores the expected transcriptional response associated with LSD1 inhibition", is interesting but the authors did not further explore/validate with experiments. It was only based on the transcriptomic data. The authors should make this part more concise. Instead, they should explore how their molecular data supports the whole in vitro data in particular the clonogenic and differentiation impact of the combination treatment which data occupied most of this manuscript.

Our molecular data fully support the cellular observations: most deregulated genes are involved in cell differentiation and genes involved in the immature status are downregulated (supporting the clonogenic data). Furthermore, cell death suggested by the transcription data was supported by the dedicated cell death assays.

Since we are convinced that restoring the expected transcriptional response to LSD1 inhibition both explains the observed cell differentiation and represents a clinically relevant effect (given ongoing clinical trials in AML), we have now validated part of our RNA-seq data using qPCR and flow cytometry in order to strengthen our conclusions (**Appendix Figure S10**).

We also reorganized the text to remove redundancy on the RNAseq data analysis: The text related to the primary samples was joined to that of the cell line model, resulting in reduction of the words by two (330 words vs 657 in the first version).

In addition, we now present data supporting that chromatin accessibility correlates with gene expression (**EV Fig.5d**), and Footprint analysis of the ATAC-seq data that indicates that CEBP and SPI1 (two pro-differentiation factors) are among the transcription factors with gain of occupancy at regulatory regions following the treatment (**EV Fig. 5c**).

Overall, the molecular data is now strengthened and supports the in vitro data observations. The text has also been reorganized to reduce redundancy.

We thank the reviewer for the comment that helped in improving the manuscript.

9th Jul 2025

Dear Dr. De Sepulveda,

Thank you for submitting your revised study, and please accept my apologies for the delay in getting back to you, which is due to the fact that one referee needed more time to provide his/her report. We have now received feedback from the three initial reviewers, and as you will see below, they are overall satisfied with the revisions. I will therefore be able to accept your manuscript when the following editorial concerns are addressed:

1/ Please address the remaining concern from referee #3.

2/ Manuscript text:

- Please remove the yellow highlighted text and only keep in track changes mode any new modification in the text.
- Please provide up to 5 keywords.
- Please remove the Conflict of interest disclosure on the title page.
- Methods:
 - o Cell lines: please indicate whether the cells were authenticated.
 - o Human samples: Include a statement confirming that informed consent was obtained from all subjects and that the experiments conformed to the principles set out in the WMA Declaration of Helsinki and the Department of Health and Human Services Belmont Report.
 - o Animals experiments: please provide the strain and origin of the mice, as well as housing and husbandry conditions. Please indicate gender and age for all experiments.
 - o Please remove "Data will be available upon reasonable request to the corresponding author." (see our policy: <https://www.embopress.org/page/journal/17574684/authorguide#availabilityofpublishedmaterial>)
 - o Statistics: please provide a statement on inclusion/exclusion criteria, sample size, blinding and randomization.
- Data Availability: Thank you for depositing your data in public repositories. Please note that data must be accessible before acceptance of the manuscript. Please remove "Materials and reagents used in the study are available upon reasonable request to the corresponding author."
- Acknowledgements: the information entered into our submission system should match the Acknowledgments section (currently, Provence-Alpes-Côte d'Azur Region, Association Cassandra, the Société Française d'Hématologie, SFH, and the EU H2020 MSCA program are not entered in the submission system).
- References: please reformat to alphabetical order, with 10 author names listed before et al.

3/ Figures:

- Please upload the main and EV figures as individual, high resolution figure files.
- Please make sure that the figures are called out in chronological order. Currently, Fig 6 is called out before Fig 5F; Table EV4 is called out before Table EV3. There are callouts for Fig 7E,F but there are no such panels, please check.
- Appendix: please add page numbers to the table of contents and remove the highlighted text in the final version.
- The EV Tables should be uploaded as separate files. Please rename them "Table EV1" - EV4 and add a legend with a short description to the top of each page (like for Table EV 4).
- During our standard figure check, we noted potential aberrations in some figure panels: please check the composition of Figure 2A, EV2A/B, Appendix Fig. S3D. Please also upload the source data for these figures. Please note that figure re-use is allowed in certain circumstances, but must be indicated in the figure legends.
- Please address the requests from our data editors:
 1. Please define the annotated p values ****/***/**/* as well as provide the exact p-values for the same in the legend of figure 7A; EV3 A, B as appropriate.
 2. Please note that the exact p values are not provided in the legends of figures 1B, C, D, E, G; 2A-F; 3B-F; 4B, C; 7B, F; EV2 C; EV4 A-C
 3. Please indicate the statistical test used for data analysis in the legends of figures 5A, F; 6A, 7A; EV3 A, B; EV5 A, C, D
 4. Please note that information related to n is missing in the legends of figures 1B, C, D, E; 2A, B, C, F; 3B-F; 4B, C; 5A, F; 6A, 7A-F; EV2 C; EV3 A, B; EV4 A-C; EV5 C.
 5. Please note that the error bars are not defined in the legends of figures 2B, C; 3B-D; 7A-F; EV2 C, EV3 A, B; EV4 A-C

4/ Source Data: Thank you for depositing your source data on BioStudies. Please rearrange them to have 1 file per figure (instead of a single file for all main figures) for easier access and download.

5/ Checklist:

- please fill in the top left corner
- please fill in the subsection "housing and husbandry conditions" in "Experimental animals"
- please fill in the full section "experimental study design and statistics"
- please fill in the subsection on human participants, authority granting ethics approval and informed consent and Helsinki

Declaration.

- please fill in the human clinical and genomic datasets subsection

6/ The paper explained: I introduced minor edits to shorten the text, please let me know if you agree, and include it in the main manuscript text:

Problem

Acute myeloid leukemia (AML) is an aggressive malignancy characterized by the accumulation of myeloid blasts in the bone marrow, which leads to hematopoiesis deficiency. AML is very heterogeneous with diverse genetic landscapes and complex intra-patient clonality that challenge treatment response, influence disease progression and ultimately affect prognosis. Standard chemotherapy is effective in fit patients, but its effects are often transient and largely insufficient for those with high-risk genetic mutations or for the elderly. Leukemic cells are more plastic than normal cells, and ultimately some cancer cells develop resistance. While several therapeutic targets have been recently identified, they have proven insufficient as monotherapies or in combination with chemotherapy.

Results

Our work focused on CDK6, a vulnerability present in several subtypes of AML that can be targeted using molecules approved for breast cancer treatment. By combining CDK6 inhibitors with existing molecules of interest in leukemia, we discovered a two-molecule regimen that restores leukemic blast differentiation and demonstrates efficacy in leukemia samples in the laboratory, including those from patients who have relapsed. In addition differentiating blasts, the combination therapy targets a small population of leukemic progenitors, and induces cell death. At the molecular level, the combination remodels chromatin accessibility, thereby impairing the imprinting of immature leukemic cells. Interestingly, unlike the single molecules, the combination induces long lasting effects on the leukemic cells. Finally, in pre-clinical mouse models, the combination therapy reduces leukemic burden in various genetically distinct primary AML samples.

Impact

Here, we demonstrate that repositioning of CDK4/6 inhibitors in AML can enhance the efficacy of drugs currently in clinical trials (LSD1 inhibitors). Furthermore, this combination may offer a viable therapeutic option in cases of resistance to conventional therapy. Finally, since the combination shows synergy, these molecules could be used at lower concentrations, potentially leading to fewer side effects. This study provides valuable insights into the potential repositioning of existing FDA-approved drugs for use in AML.

7/ Synopsis:

I introduced minor edits in your text, please let me know if you agree or amend as you see fit:

"Several novel protein targets have shown promising preclinical potential for acute myeloid leukemia, but have provided limited benefit for patients. This prompted the investigation of combinations of clinically available molecules.

- Combined inhibition of CDK6 and LSD1 remodels chromatin accessibility, notably at the key myeloid transcription factors CEBP and SPI1 binding sites.
- The combinatorial approach fully restores the expected transcription program associated with LSD1 inhibition.
- Dual CDK6 and LSD1 inhibition synergistically promotes myeloid differentiation of all tested AML samples.
- This therapeutic combination elicits sustained effects on leukemic progenitor cells, opening new avenues for drug repositioning in AML treatment."

Please upload the synopsis as a separate file (jpeg, png or tiff file 550 px wide x 300-600 px high).

8/ As part of the EMBO Publications transparent editorial process initiative (see our Editorial at <http://embomolmed.embopress.org/content/2/9/329>), EMBO Molecular Medicine will publish online a Review Process File (RPF) to accompany accepted manuscripts.

This file will be published in conjunction with your paper and will include the anonymous referee reports, your point-by-point response and all pertinent correspondence relating to the manuscript. Let us know whether you agree with the publication of the RPF and as here, if you want to remove or not any figures from it prior to publication.

I look forward to receiving your revised manuscript.

Yours sincerely,

Lise Roth

**** Reviewer's comments ****

Referee #1 (Remarks for Author):

I am happy with the new version and would accept this version for publication.

Referee #2 (Comments on Novelty/Model System for Author):

The experiments performed are reasonable but it is difficult to know if the findings here will prompt clinical translation.

Referee #2 (Remarks for Author):

The authors have addressed my previous comments. The in vivo experiments show a modest improvement in survival and thus may not prompt clinical translation. Perhaps future studies can build on this.

Referee #3 (Remarks for Author):

The authors have addressed most of the points satisfactorily raised by this reviewer. The manuscript is now solid, and interesting and will have a good impact on the AML field.

Concerning the point raised about the colony assay, despite the justifications used by the authors, maybe the authors were/are not aware that most of the labs that work on primary AMLs do screen their samples for their leukemic clonogenic capacity and verify that the colonies in their studies were leukemic (despite not reporting as such in their publications); everyone knows in the field that this assay is not reliable but unfortunately no one has bothered to report it apart from PMID: 31936647. That said, since using normal CD34+ yielded little effect, one would assume that the impact was really from the leukemic cells. The authors still need to make a comment about the potential caveat (for not having screened their AMLs) in the discussion.

Manuscript EMM-2024-20757

Point-by-point responses; responses in blue

1/ Please address the remaining concern from referee #3.

“The authors still need to make a comment about the potential caveat (for not having screened their AMLs) in the discussion.”

To take into account the reviewer’s comment, we have added the following paragraph to the discussion: “We investigated the impact of combined CDK6 and LSD1 inhibition on progenitor cells from primary AML samples using colony assays. The observed reduction in colony formation is likely attributed to the targeting of leukemic progenitors, supported by the high blast percentage in the patient samples and the lack of inhibitory effect on normal CD34⁺ hematopoietic cells. However, the leukemic origin of the colonies was not directly confirmed through analysis of the colonies themselves.”

4th Aug 2025

Dear Dr. De Sepulveda,

Thank you for sending your revised files and dealing with the remaining source data issues. I am pleased to inform you that your manuscript is accepted for publication and is now being sent to our publisher to be included in the next available issue of EMBO Molecular Medicine!

Please note that I have added the URL to the deposited Source Data in the manuscript text file:
(<https://www.ebi.ac.uk/biostudies/studies/S-BSST2077?query=S-BSST2077>).

With kind regards,

Lise
